# Improving Out-of-Distribution Detection via Dynamic Covariance Calibration

Kaiyu Guo [1 2]   Zijian Wang [1]   Tan Pan [3 2]   Brian C. Lovell [1]   Mahsa Baktashmotlagh [1]

## Abstract

Out-of-Distribution (OOD) detection is essential for the trustworthiness of AI systems. Methods using prior information (i.e., subspace-based methods) have shown effective performance by extracting information geometry to detect OOD data with a more appropriate distance metric. However, these methods fail to address the geometry distorted by ill-distributed samples, due to the limitation of statically extracting information geometry from the training distribution. In this paper, we argue that the influence of ill-distributed samples can be corrected by dynamically adjusting the prior geometry in response to new data. Based on this insight, we propose a novel approach that dynamically updates the prior covariance matrix using real-time input features, refining its information. Specifically, we reduce the covariance along the direction of real-time input features and constrain adjustments to the residual space, thus preserving essential data characteristics and avoiding effects on unintended directions in the principal space. We evaluate our method on two pre-trained models for the CIFAR dataset and five pre-trained models for ImageNet-1k, including the self-supervised DINO model. Extensive experiments demonstrate that our approach significantly enhances OOD detection across various models. The code is released at https://github.com/workerbcd/ooddcc.

## 1. Introduction

Deep learning has achieved remarkable success in pattern recognition tasks (He et al., 2016; Han et al., 2022). How-

[1]School of Electrical Engineering and Computer Science, University of Queensland, Brisbane, Australia [2]Shanghai Academy of AI for Science (SAIS), Shanghai, China [3]Artificial Intelligence Innovation and Incubation Institute, Fudan University, Shanghai, China. Correspondence to: Mahsa Baktashmotlagh <m.baktashmotlagh@uq.edu.au>.

*Proceedings of the 42nd International Conference on Machine Learning*, Vancouver, Canada. PMLR 267, 2025. Copyright 2025 by the author(s).

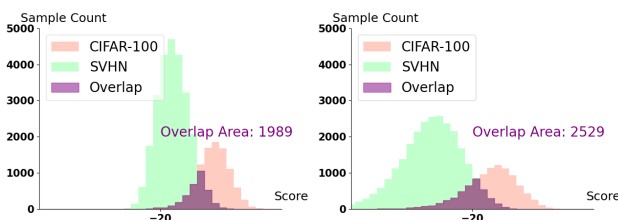

|(a) with dynamic adjustment | (b) without dynamic adjustment |

*Figure 1.* Visualization of Mahalanobis distance score distributions with (w) and without (w/o) Dynamic Adjustment. We compare scores from the CIFAR-100 dataset (ID) and the SVHN dataset (OOD) using a CIFAR-100 pre-trained DenseNet. Without dynamic adjustment, the scores have high variance and a large overlapping region. With dynamic adjustment, the variance is lower and the overlap is reduced.

ever, deep models are often well-fitted to the data they are trained on, known as the in-distribution (ID) data, and may perform poorly on unseen or novel data, referred to as out-of-distribution (OOD) data. Recognizing OOD samples is crucial for ensuring the safety and reliability of AI systems, especially in critical applications like autonomous driving (Filos et al., 2020), medical diagnostics (Fu et al., 2023), and surveillance (Idrees et al., 2018). Therefore, OOD detection is proposed to separate ID and OOD data to define the performance boundaries of deep learning models.

The goal of OOD detection is to define a score function $s(\cdot)$ that assigns higher scores to ID data and lower scores to OOD data. Using a threshold $\lambda$, we can classify inputs as:

$$D(x, s, \lambda) = \begin{cases} x \in ID & \text{if } s(x) > \lambda \\ x \in OOD & \text{if } s(x) < \lambda \end{cases}$$

A variety of score functions have been proposed for OOD detection (Yang et al., 2024). In this paper, we focus on designing score functions for test-time OOD detection, where models do not require retraining with ID data.

With the widespread use of pre-trained models and contrastive learning, classifier-free methods for test-time OOD detection have been proposed, among which measuring the distance between features is an effective approach. Previous works (Sehwag et al., 2021; Lee et al., 2018) have shown

the effectiveness of the Mahalanobis distance in OOD detection. Mahalanobis distance computes distances using a covariance matrix derived from the ID data, capturing its information geometry. However, the distortion of information geometry caused by the outlier features in the training data is neglected in these methods. Specifically, if the ID data has high variance deviating from semantic directions due to outliers, the Mahalanobis distance may become less sensitive to OOD samples that align with these directions. This can lead to misclassifying OOD data as ID.

Recently, methods that project features onto the subspace of ID data (Wang et al., 2022; Ammar et al., 2024; Chen et al., 2023) have been proposed, which can be viewed as matrix-induced distance scores. These approaches can be seen as attempts to mitigate this issue by replacing the covariance matrix in the Mahalanobis distance with tailored matrices. By eliminating components associated with extreme variances, these methods aim to reduce the influence of outliers. However, this may also remove important information in the ID distribution and does not provide a targeted adjustment for specific OOD directions. This raises the question: How can we reduce the effect of outlier features on the information geometry without losing important characteristics of the ID data?

To address this challenge, we propose a novel approach that dynamically adjusts the prior matrix using real-time input features. Our key insight is that real-time input features capture local information about the unknown distribution, which can be used to refine the prior information geometry. Specifically, we propose to adjust the covariance matrix by decreasing the covariance in the direction of the real-time features. This effectively contracts the covariance structure along targeted directions, making the distance measure more sensitive to deviations in those directions and improving OOD detection performance.

However, not all information in the real-time features is suitable for adjustment. To avoid disrupting the main structure of the ID data, we restrict changes to the residual space. By adjusting the covariance in the residual space, we preserve the main patterns of the ID data while enhancing OOD detection by increasing sensitivity to samples that deviate in these less significant directions. This focus also reduces noise and redundancy, improving the distinction between ID and OOD data. Our OOD detection score function is computed on the transformed features, which allows the ID distribution to form a denser manifold. Figure 1a shows the effectiveness of our method. With dynamic adjustment, the scores have lower variance and a smaller overlapping region between ID and OOD data, indicating improved separability. In summary, Our main contributions are as follows:

- We introduce a new perspective for designing distance-based OOD score functions by utilizing real-time input

features to dynamically adjust the prior geometry. Our proposed OOD detection score function reduces redundant information in the covariance matrix in real time, making it better tailored to OOD detection. By restricting adjustments to the residual space, we enhance sensitivity to OOD samples while preserving the essential structure of the ID data.

- Our regularization technique method can be applied to distance measures that rely on covariance structures. This broadens the applicability of our approach to various problems where adjusting the sensitivity of distance measures can improve performance.

- We evaluate our method on CIFAR benchmarks with 2 pre-trained models and on Imagenet with 5 pre-trained models including the self-supervised pre-trained model DINO, showing it outperforms state-of-the-art (SOTA) in most cases.

## 2. Related Work

### 2.1. Out-of-distribution Detection

With the observation that the deep learning models are over-confident in classifying samples over different semantic distributions, out-of-distribution detection emerges to reject the samples with different semantic information from the training data (Yang et al., 2024). Unlike outlier detection methods (Sehwag et al., 2021), OOD detection aims to determine whether real-time data belongs to the in-distribution or out-of-distribution. Thus, the group information can not be obtained from the unseen distribution.

Many methods develop new training strategies (DeVries & Taylor, 2018; Wang et al., 2021; Vyas et al., 2018; Chen et al., 2021) or utilize the data augmentation (Hein et al., 2019; Hendrycks et al., 2022) to manipulate the confidence. However, this kind of method requires a training phase, which may take a long time to achieve a score function with different deep learning models. Moreover, the computational resource is also a limitation of these methods. So, in this paper, we focus on the test-time OOD detection task which consumes less time to get the score function.

### 2.2. Test-time OOD detection

As aforementioned in the introduction, test-time OOD detection aims to design the OOD score $D(x, s, \lambda)$ with a fixed pre-trained model. According to (Yang et al., 2024; Ammar et al., 2024), the test-time OOD detection method can be mainly divided into five categories: logit-based, gradient-based, feature-based, distance-based, density-based, and subspace-based. For the logit-based methods (Hendrycks & Gimpel, 2017; Liu et al., 2020; Basart et al., 2022), these works tend to detect the OOD data based on the high confi-

dence lying in the logit or probability from the ID data. For gradient-based methods (Huang et al., 2021; Behpour et al., 2023; ElAraby et al., 2023), these method aims to detect OOD data with the differences of gradients. The gradient-based method, GradNorm, has been proven to be related to the logits and feature norms (Huang et al., 2021). For the feature-based methods, feature manipulations like clipping (Sun et al., 2021) and scaling (Xu et al., 2024) are proposed to remove some redundant information. Similarly, the weight pruning methods (Sun & Li, 2022; Ahn et al., 2023) also tend to select important neurons with this motivation. In addition, weight pruning methods are the same as feature pruning, where different masks are given to the features corresponding to different logits. For density-based methods, these methods (Morteza & Li, 2022; Peng et al., 2024) design the OOD score function by modeling the density of a certain distribution. Distance-based methods (Basart et al., 2022; Lee et al., 2018; Liu & Qin, 2023) utilize the property that the OOD features reside in the outlier of the feature clusters. However, these methods require high-quality features and may result in large variances with low-quality features. For subspace methods (Wang et al., 2022; Ammar et al., 2024; Chen et al., 2023), these methods project the real-time features to a subspace of ID data, and the scores are designed based on norms or distances. So, these methods can be regarded as distance-based score functions where the distances are matrix-induced. In this paper, these score functions are highly related to the matrix-induced distance scores. Thus, the design of the subspace methods can be viewed as defining new distances with various prior matrices. Unlike previous studies that only estimate the prior matrix on the training samples, we dynamically estimate the matrix on the geometry of training distribution and local geometry of unknown distribution from real-time data.

## 3. Preliminary and Insight

Given $\mathcal{X}$ as data space and $\mathcal{Y}$ as the label space, the ID and OOD distributions can be defined as joint distributions $P_{X_{in}Y_{in}}$ over $\mathcal{X}_{id} \times \mathcal{Y}_{id}$, and $P_{X_{ood}Y_{ood}}$ over $\mathcal{X}_{ood} \times \mathcal{Y}_{ood}$, respectively. Given a pre-trained model $h \circ g$, where $h$ is an encoder and $g$ is a classifier, we obtain the ID feature distribution $\mathcal{F}_{id} = h(\mathcal{X}_{id})$ and the OOD feature distribution $\mathcal{F}_{ood} = h(\mathcal{X}_{ood})$. Considering an anchor feature $f_a \in \mathcal{F}_{id}$, an intuitive insight is that $d(f_{id}, f_a) \leq d(f_{ood}, f_a)$, where $d(\cdot, \cdot)$ is a pre-defined distance metric, with $f_{id} \in \mathcal{F}_{id}$ and $f_{ood} \in \mathcal{F}_{ood}$. Previous work (Liu & Qin, 2023) has revealed the shortcomings of using Euclidean distance with global mean $\mu$ as an anchor point in OOD detection. However, the Mahalanobis distance has been demonstrated to be effective as a detection score (Lee et al., 2018; Sehwag et al., 2021), highlighting the necessity of considering information geometry when designing distance-based scores. Therefore, we simplify the distance designed for OOD detection scores as

a matrix-induced distance:

$$d_M(f, f_a) = \sqrt{(f - f_a)^\top M (f - f_a)}, \qquad (1)$$

where the matrix $M$ defines the geometry and $f$ is a real-time input feature.

For the Mahalanobis distance OOD detection score (Lee et al., 2018; Sehwag et al., 2021), the matrix $M$ is the precision matrix $\Sigma^{-1}$ for the distribution of each class. These methods consider each class of ID data as a distribution, and the score is defined as the minimum distance from the data point to the mean point over the geometry of each class.

For residual space method (Wang et al., 2022), $M$ is defined as $R^\top R$, where $R$ is the basis of the residual space of the ID data. Since the score function is $\sqrt{(f - \mu)^\top R^\top R (f - \mu)}$, this method effectively measures the distance between the data point and the mean point over the manifold of distribution $\mathcal{N}(\mu, (R^\top R)^{-1})$

Similarly, for the principal space method (Ammar et al., 2024), $R$ is the basis in the principal space of the data. As the features are standardized in the first stage, the score considers the distance between the data and the zero point over the manifold $\mathcal{N}(0, (P^\top P)^{-1})$, where $P$ is the basis of the principle space of ID data. The dominator in the score function can be viewed as normalization for each feature.

As discussed above, previous works tend to consider the static geometry from ID data. Therefore, outlier features from the training samples can consistently affect the distance for every real-time input data. To mitigate this issue using the local information from real-time features, the distance in Equation 1 can be transformed as follows:

$$d_\mathcal{M}(f, f_a) = \sqrt{(f - f_a)^\top \mathcal{M}(f)(f - f_a)} \qquad (2)$$

where $f$ is the real-input feature, $f_a$ is the anchor feature, and $\mathcal{M} : \mathbf{R}^d \to \mathbf{R}^{d \times d}$ maps the $d$ dimension features to the $d \times d$-matrix space. Figure 2 shows the difference between the Euclidean distance and the matrix-induced distance as defined in equations 1 and 2. In the following section, we introduce our design for the matrix $\mathcal{M}$ in Equation 2.

## 4. Methodology

In Section 4.1, we propose a method to mitigate the impact of outlier features in the covariance matrix of the Mahalanobis distance, with the core objective of finding a projection matrix $\mathcal{M}$ that maximally separates ID and OOD samples. We hypothesize that these outlier features may align with the features from novel distributions. We also present the condition to ensure $\mathcal{M}$ to induce a valid distance. In Section 4.2, we suggest constraining adjustments to the residual space. We argue that, within the principal space of

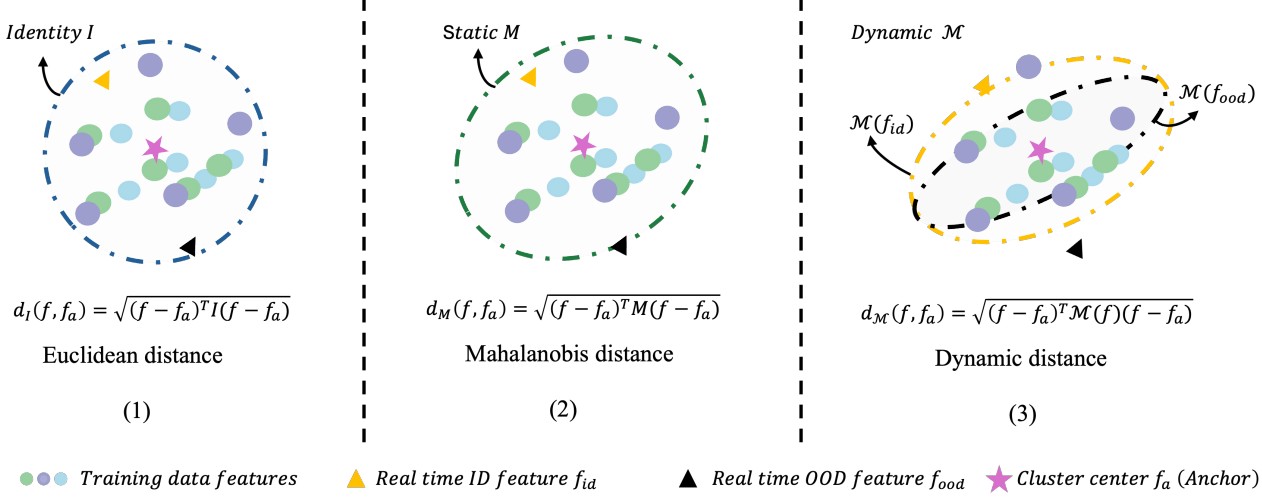

$$d_I(f, f_a) = \sqrt{(f - f_a)^T I (f - f_a)}$$

Euclidean distance

(1)

$$d_M(f, f_a) = \sqrt{(f - f_a)^T M (f - f_a)}$$

Mahalanobis distance

(2)

$$d_{\mathcal{M}}(f, f_a) = \sqrt{(f - f_a)^T \mathcal{M}(f)(f - f_a)}$$

Dynamic distance

(3)

● ● ● *Training data features*     ▲ *Real time ID feature $f_{id}$*     ▲ *Real time OOD feature $f_{ood}$*     ★ *Cluster center $f_a$ (Anchor)*

*Figure 2.* Visualization of the difference among distances calculated with no prior matrix, a static prior matrix, and a dynamic prior matrix. The distance with no prior matrix is Euclidean distance where $M$ is an identity matrix. In the case of the static prior matrix, the geometry can not be adjusted, even if it is distorted by outlier features in the training samples. With the dynamic prior matrix, the geometry can be refined using the local information from the real-time input features.

the training distribution, there exists information that cannot be shrunk, such as semantic information or model distribution. Based on these findings, in Section 4.3, we propose an OOD detection score and illustrate how its design meets the properties of a valid distance function induced by $\mathcal{M}$.

### 4.1. Dynamic Matrix Estimation

Here, we aim to design a mapping $\mathcal{M}$ to better separate ID and OOD samples. As discussed in (Pinele et al., 2019), the Mahalanobis distance measures distances within the space of Gaussian distributions, influenced by their covariance structures. Therefore, we can consider the covariance matrix $\Sigma$ as a key matrix encoding the primary geometric characteristics of in-distribution (ID) data under Gaussian assumptions. When a feature $f$ from a novel distribution feeds in, we aim to reduce its influence within the covariance $\Sigma$. To achieve this, we modify the covariance matrix to $\Sigma - ff^\top$. The following proposition explains this property mathematically.

**Proposition 4.1.** *Given vector $v$, $v^\top(\Sigma - ff^\top)v \leq v^\top \Sigma v$. The difference can be determined by the square of similarity $(v^\top f)^2$.*

To enhance the Mahalanobis distance for OOD detection, given a feature $f$ from a new distribution, we define the mapping $\mathcal{M}(f) = (\Sigma - ff^\top)^{-1}$. However, with $f$ fixed, $\mathcal{M}(f)$ may not be a positive definite matrix. Fortunately, we find that with a given anchor point $a \neq 0$, the quadratic form $(f - a)^\top \mathcal{M}(f)(f - a)$ can be guaranteed to have a positive value under certain conditions and introduce a theorem to provide this condition.

**Theorem 4.2.** *Given a feature $f \in \mathbf{R}^d$, a non-zero feature $a \in \mathbf{R}^d$, and a symmetric positive definite matrix $\Sigma \in \mathbf{R}^{d \times d}$, we define $p = f^\top \Sigma^{-1} f$, $q = a^\top \Sigma^{-1} a$, and $s = f^\top \Sigma^{-1} a$. Setting $d(f) = (f - a)^\top(\Sigma - ff^\top)^{-1}(f - a)$, if $p < 1$, then $d(f) \geq 0$; if $p > 1$ and $(s-1)^2 \leq (p-1)(q-1)$, then $d(f) \geq 0$.*

For OOD features, we can not estimate the magnitude of $p$. Therefore, to maintain $d(f)$ positive, we ensure that $(s-1)^2 \leq (p-1)(q-1)$. Interestingly, under the condition where $p \gg 1$ and $q \gg 1$, this inequality simplifies to $(s - 1)^2 \leq pq$, a condition satisfied by the Cauchy-Schwarz inequality when $s \geq 0$[1]. In Section 4.3, we will discuss why this condition can be satisfied.

### 4.2. Projection in Residual Space

As mentioned earlier, our foundational design of $\mathcal{M}$ involves subtracting $ff^\top$. However, this operation may affect the principal information of the training distribution, which could align with real-time ID features. Furthermore, studies in compatible learning (Pan et al., 2023) have illustrated that features from different pre-trained models can come form different distributions. We also visualize this phenomenon in Figure 3 with ImageNet-1k (Deng et al., 2009) pre-trained ViT (Dosovitskiy et al., 2021) and DeiT (Touvron et al., 2021) models. The separated clusters demonstrate that model distributions are principal factors affecting feature distributions. Consequently, shared model distributions may influence the direction of adjustment. To preserve semantic information while minimizing the impact

---

[1]When $s < 0.5$, the inequality is obviously satisfied.

of shared model information on adjustment direction with real-time features, we confine the adjustment to the residual space (Wang et al., 2022) of the training distribution.

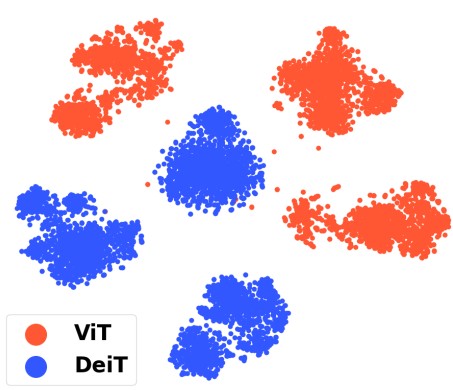

*Figure 3.* Visualization of the model distribution between ImageNet-1k pre-trained ViT and DeiT. We labeled the features from ViT with orange colour and DeiT with blue colour. The features from different models are distributed separately, which suggests the model distribution is also the principal information affecting the feature distribution.

In this context, the feature from the unknown distribution in the penultimate layer can be represented as $f_r = \sum_i a_i b_i = \mathbf{a}^\top \mathbf{B}$, where $\mathbf{B} = \{b_i\}$ is the basis matrix of the residual space, and $\mathbf{a} = \{a_i\}$ is the coefficient vector. Consequently, $\mathcal{M}(f)$ is defined as $\mathcal{M}(f) = (\Sigma - \mathbf{B}^\top \mathbf{a} \mathbf{a}^\top \mathbf{B})^{-1}$. According to (Wang et al., 2022), $\mathbf{B}$ is the eigenvectors corresponding to the smallest eigenvalues of $\Sigma$.

VIM research (Wang et al., 2022) suggests that OOD features typically exhibit more energy in residual space. Moreover, Neco (Ammar et al., 2024) builds on the Neural Collapse phenomenon (Papyan et al., 2020) by suggesting that features in the penultimate layer of ID and OOD data can become increasingly orthogonal. This support the assumption that the OOD features predominantly occupy the null space of ID data, which also aligns with observations in VIM. Further enhancing this, we assume that the stronger collapse phenomenon may allow the residual space to capture more out-of-distribution information from OOD features. To strengthen the collapse phenomenon, all features undergo $L_2$ normalization in the initial stage (Haas et al., 2023).

### 4.3. Dynamic Covariance for Enhanced OOD Score

Building upon the theoretical findings from the previous sections, we aim to define a score function that achieves our ultimate goal of effectively separating ID and OOD samples. To this end, by utilizing deviation features derived from each class mean $\mu_c$, we construct a zero-mean distribution and adjust the within-class covariance matrix accordingly.

Specifically, from the mean of each class $\mu_i$, we can derive a deviation feature in the training sample as $r_{in} = f_{in} - \mu_c$, where $c$ is the class label to which $f_{in}$ belongs. Then, we have $R = \{r_{in}\}$ forming a zero-mean distribution, and we define $\mathcal{M}(f) = (\Sigma_R - \mathbf{B}^\top \mathbf{a} \mathbf{a}^\top \mathbf{B})^{-1}$, where $\Sigma_R$ is the covaraince matrix of $R$, namely the within-class covariance of $\{f_{in}\}$. Given the real-time feature $f$, the score function can be defined as:

$$s(f) = -\min_i \sqrt{r_i^\top \mathcal{M}(f) r_i}, r_i = f - \mu_i, i = 0, 1, \cdots, N_c \tag{3}$$

Where $N_c$ is the number of classes. The function $s(z)$ aims to find the minimum distance between the real-time feature and different class means, calculated over the dynamically adjusted geometry based on the within-class covariance matrix $\Sigma_R$.

Rethinking the question in Section 4.1, if we consider the $\Sigma$ in Theorem 4.2 as the within-class covariance $\Sigma_R$ and $a$ as one class mean $\mu_c$, we can easily obtain $p \gg 1$ and $q \gg 1$, since $\Sigma_R$ reflects the compactness of the class clusters and none of the features tends to cluster around zero point. As $f$ and $\mu_c$ mostly lie in the span of the positive eigenspace of $\Sigma_R$, thus $s \geq 0$. Therefore, we can build a distance with $\mathcal{M}$. Similarly, the same property can also be preserved when $f$ in the residual space as $f_r$ and $a$ as $\mu_c - f_r'$, where $f = f_r + f_r'$. We visualize the distributions of $p$ and $q$ in the Section 5.7 and $s$ in the appendix.

## 5. Experiments

### 5.1. Experiment Setup

To validate the effectiveness of our method, we conduct experiments on two standard out-of-distribution (OOD) detection benchmarks. In the first benchmark, we use the official test split of CIFAR-10/CIFAR-100 as the in-distribution (ID) datasets, with six datasets serving as OOD data: SVHN (Netzer et al., 2011), Textures (Cimpoi et al., 2014), Places365 (Zhou et al., 2017), LSUN (Yu et al., 2015), LSUN_Resize (Yu et al., 2015) and iSUN (Xu et al., 2015).

For the second benchmark, we follow (Huang & Li, 2021) to adopt ImageNet-1k (Deng et al., 2009) as the ID dataset. There are six OOD datasets selected, including Texture (Cimpoi et al., 2014), SUN (Xiao et al., 2010), Places (Zhou et al., 2017), iNaturalist (Van Horn et al., 2018), Imagenet-O (Hendrycks et al., 2021) and OpenImage-O (Wang et al., 2022). Particularly, we use the subset of SUN, Places and Texture curated by Mos (Huang & Li, 2021) as the OOD dataset. The OpenImage-O is a subset of OpenImage (Krasin et al., 2017) selected by VIM (Wang

*Table 1.* Comparision with different post-hoc OOD detection methods on **CIFAR** benchmarks. We present the AUROC and FPR95 results on DenseNet and WideResNet and the average results over 2 ID datasets. The results of CIFAR-10/CIFAR-100 are averaged over 6 OOD datasets. The detailed results can be viewed in the appendix.

| Method | DenseNet | | | | | | WideResNet | | | | | |
| --- | --- | --- | --- | --- | --- | --- | --- | --- | --- | --- | --- | --- |
| | CIFAR-10 | | CIFAR-100 | | Avg. | | CIFAR-10 | | CIFAR-100 | | Avg. | |
| | AUROC↑ | FPR95↓ | AUROC↑ | FPR95↓ | AUROC↑ | FPR95↓ | AUROC↑ | FPR95↓ | AUROC↑ | FPR95↓ | AUROC↑ | FPR95↓ |
| MSP (Hendrycks & Gimpel, 2017) | 92.5 | 48.72 | 74.37 | 80.13 | 83.44 | 64.43 | 91.07 | 50.64 | 76.93 | 77.48 | 84 | 64.06 |
| Energy (Liu et al., 2020) | 94.65 | 26.2 | 81.17 | 68.44 | 87.91 | 47.32 | 91.86 | 33.74 | 79.83 | 71.95 | 85.85 | 52.85 |
| maxLogit (Basart et al., 2022) | 94.64 | 26.36 | 81.06 | 68.53 | 87.85 | 47.45 | 91.84 | 33.61 | 79.92 | 72.37 | 85.88 | 52.99 |
| ODIN (Liang et al., 2018) | 94.65 | 26.35 | 81.06 | 68.53 | 87.86 | 47.44 | 91.85 | 33.62 | 79.93 | 72.38 | 85.89 | 53 |
| Mahalanobis (Lee et al., 2018) | 85.9 | 47.64 | 77.56 | 58.08 | 81.73 | 52.86 | 90.88 | 47.58 | 79.35 | 59.63 | 85.12 | 53.61 |
| GEM (Morteza & Li, 2022) | 88.01 | 31.73 | 84.19 | 56.93 | 86.1 | 44.33 | 93.22 | 37.28 | 82.71 | 57.15 | 87.97 | 47.22 |
| KNN (Sun et al., 2022) | 96.79 | 16.16 | 87.56 | 42.3 | 92.18 | 29.23 | 93.68 | 33.56 | 86.34 | 48.32 | 90.01 | 40.94 |
| ReAct (Sun et al., 2021) | 95.76 | 23.59 | 82.98 | 67.38 | 89.37 | 45.49 | 92.09 | 34.06 | 80.69 | 72.26 | 86.39 | 53.16 |
| Line (Ahn et al., 2023) | 96.99 | 14.75 | 88.76 | 35.11 | 92.88 | 24.93 | 78.94 | 61.6 | 66.33 | 83.45 | 72.64 | 72.53 |
| DICE (Sun & Li, 2022) | 95.01 | 21.44 | 86.55 | 51.66 | 90.78 | 36.55 | 90.48 | 34.44 | 78.44 | 71.04 | 84.46 | 52.74 |
| FDBD (Liu & Qin, 2023) | **97.23** | **13.86** | 89.25 | 50.57 | 93.24 | 32.22 | 92.27 | 36.87 | 85.14 | 65.77 | 88.71 | 51.32 |
| ours | 96.83 | 14.63 | **92.38** | **29.98** | **94.61** | **22.31** | **96.18** | **18.57** | **89.08** | **44.89** | **92.63** | **31.73** |

et al., 2022). There are no overlapping categories between the OOD datasets and the ID dataset. We consider *FPR95* and *AUROC* as performance metrics across both benchmarks. Please refer to the appendix material for a more detailed explanation.

## 5.2. Evaluation on CIFAR benchmark

We compare our method with a diverse set of post-hoc OOD detection approaches, including one probability-based method (MSP (Hendrycks & Gimpel, 2017)), three logit-based methods (Energy (Liu et al., 2020), maxLogit (Sun et al., 2022), ODIN (Liang et al., 2018)), one density-based method (GEM (Morteza & Li, 2022)), and three distance-based methods (Mahalanobis (Lee et al., 2018), KNN (Sun et al., 2022), FDBD (Liu & Qin, 2023)). Additionally, we compare our method with feature clipping and weight pruning techniques such as ReAct (Sun et al., 2021), DICE (Sun & Li, 2022), and LINE (Ahn et al., 2023).

Table 1 presents the averaged AUROC and FPR95 metrics over six OOD datasets for the CIFAR benchmarks. Our method demonstrates the best AUROC and FPR95 results on average among all comparison methods. In particular, our proposed method reduces the average FPR95 by 9.9% on DenseNet and by 19.6% on WideResNet compared to the second-best method, FDBD.

FDBD is the most relevant comparison method, as it detects outlier data by employing a distance-based score function. However, FDBD's effectiveness depends heavily on the quality of the extracted features, specifically the degree of inter-class separability. This sensitivity can be observed by comparing results on CIFAR-10 and CIFAR-100: when we decrease the class separability by increasing the number of classes, the performance of FDBD is significantly impacted, even with the same backbone model. From this, we can infer that FDBD performs better in scenarios where training features are well-clustered and not prone to poor distribution. These conditions are restrictive and difficult to achieve in

general settings, such as with a high number of classes in the training set or limitations in the model's capacity. Our method addresses this limitation by regularizing prior geometry with real-time features, enabling it to achieve a greater performance improvement over FDBD on CIFAR-100. This phenomenon shows that our method is more robust under complex environments than FDBD.

## 5.3. Evaluation on ImageNet Benchmark

In this setting, we compare our method with baseline methods (MSP (Hendrycks & Gimpel, 2017), Energy (Liu et al., 2020), ODIN (Liang et al., 2018), React (Sun et al., 2021) and maxLogit (Basart et al., 2022)), distance-based methods (Malahnobis distance (Lee et al., 2018), KL divergence (Basart et al., 2022), KNN (Sun et al., 2022) and FDBD (Liu & Qin, 2023)) and subspace methods (VIM (Wang et al., 2022), Neco (Ammar et al., 2024) and WDiscOOD (Chen et al., 2023))

Table 2 comparative analysis of various OOD detection methods on the ImageNet benchmark. The results show that our method achieves state-of-the-art performance on both AUROC and FPR95 metrics across all four pre-trained models, highlighting its robustness and adaptability in large-scale OOD detection tasks. Due to NaN scores encountered with the SwinV2-B/16 model, the reported WDiscOOD results are based on three models (*i.e.*, ViT, ResNet-50, and DeiT). More detailed results can be viewed in the appendix.

As mentioned in (Wang et al., 2022), the residual space method is limited by the feature quality of the original network. Moreover, we find that all subspace methods (Wang et al., 2022; Ammar et al., 2024; Chen et al., 2023) meet this problem. This phenomenon can be demonstrated by the weak performance of the subspace methods on ResNet-50 from Table 2. Instead of statically relying on the prior estimated geometry from training samples, our method dynamically refines the geometry to reduce the effect of ill-distributed samples in training distribution. Thus, our

method achieves significantly better performance compared to subspace methods on ResNet-50.

## 5.4. Evaluation using DINO

DINO (Caron et al., 2021) is a classical self-supervised pre-training method. The DINO pre-trained ViT model is publicly available, and we follow the same evaluation protocol as in the ImageNet benchmark. Due to the limitations inherent in self-supervised learning, we restrict our comparisons to methods that do not require linear classifiers.

As shown in Table 3, our method achieves SOTA performance on DINO in this setting. Notably, we omit WDiscOOD results because the whitening transformation in WDiscOOD produced *NaN* values with DINO features, making the evaluation infeasible. Our approach excels in handling the unique characteristics of self-supervised features, achieving superior OOD detection performance. This highlights our method's adaptability and effectiveness in scenarios involving self-supervised pre-trained models.

## 5.5. Ablation Study

We show the unique functionality of our method by studying three key components of our method: 1) **DME**: Dynamic Matrix Estimation; 2) **RSP**: Residual Space Projection for the real-time adjustment; 3) **DCM**: Dynamic Covariance Modeling with within-class covariance. We conducted an ablation study of both CIFAR and ImageNet benchmarks. DenseNet is selected as the backbone network for CIFAR, while ViT and ResNet-50 are exploited as the backbone for ImageNet. This setup provides holistic coverage of a small-scale to a large-scale dataset, and transformer to CNN. We report the FPR95 index in this ablation study in Table 4. The baseline method is the Mahalanobis distance with normalized features.

As shown in the table, the method without considering the real-time features can not perform better than the full method in all scenarios, illustrating the effectiveness of our dynamic estimation of $\mathcal{M}$. In addition, we can find that the DCM significantly contributes to the final performance for some backbone models, such as DenseNet and ViT. We assume the large between-class covariance distorts the geometry with which the data can not form a dense manifold. We discuss the benefit of within-class covariance in the appendix. Also, without RSP, the performance can deteriorate in some situations like ResNet-50, which shows the importance of avoiding prior adjustment in the principal space. Note that RSP can only be computed using the full covariance matrix (within-class + between-class), differing from the RSP in the full model, which specifically optimizes the within-class covariance. RSP can be detrimental in this situation due to the large between-class covariance. But it can solely work on poorly clustered scenarios since

within-class covariance dominates the overall covariance matrix. However, DCM may have an inverse effect, as the poorly clustered features could result in a smoother and more unified distribution.

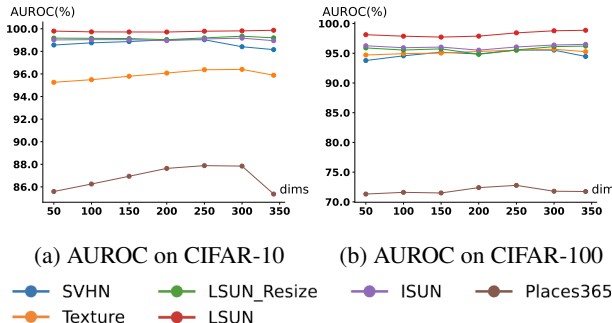

(a) AUROC on CIFAR-10     (b) AUROC on CIFAR-100

*Figure 4.* AUROC on CIFAR-10 and CIFAR-100 pre-trained DenseNet with different residual space dimension

## 5.6. Performance w.r.t Residual Dimensionality

Figure 4 and 5 show how the performance varies with different dimensions of the residual spaces on four pre-trained models: CIFAR-10 and CIFAR-100 pre-trained DenseNet, ImageNet-1k pre-trained ViT and ResNet50. As shown in Figure 4, the performance does not fluctuate much with different dimensions of residual space. In addition, the best performance on both pre-trained models is achieved at dimensionalities around 250, indicating the consistency of our method on the same model. The stable trend at the start in Figure 5 shows the robustness of residual space in large-scale OOD detection. The trend in the ViT experiments is consistent with the findings in VIM (Wang et al., 2022).

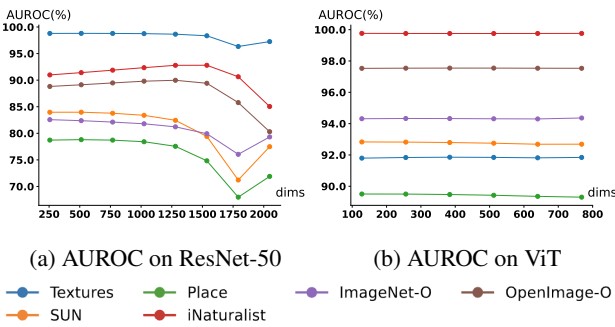

(a) AUROC on ResNet-50     (b) AUROC on ViT

*Figure 5.* AUROC on ImageNet-1k pre-trained ResNet and ViT with different residual space dimensions

## 5.7. Analyzing $p$ and $q$ Values

Theorem 4.2 emphasizes that $p$ and $q$ should be much larger than 1 to guarantee the solid distance. Since the prior adjustments are confined to the residual space, $p = f_r^\top \Sigma_R^{-1} f_r$, where $f_r = \mathbf{a}^\top \mathbf{B}$ is the real-time feature restricted in residual space, and $q = (\mu_c - f_r')^\top \Sigma_R^{-1}(\mu_c - f_r')$. $\mu_c$ is selected

*Table 2.* Comparision with different post-hoc OOD detection methods on **ImageNet-1k** benchmark. We present the AUROC and FPR95 results on ViT, ResNet-50, SwinV2-B, and DeiT. We also provide the average results over the 4 pre-trained models. The results of the four pre-pretrained models are averaged over 6 OOD datasets. The detailed results can be viewed in the appendix. As we can not achieve solid results with WDiscOOD on ImageNet-1k pre-trained SwinV2-B/16, the average results are from the other 3 pre-trained models.

| Method | Models | | | | | | | | | |
|---|---|---|---|---|---|---|---|---|---|---|
| | ViT | | ResNet-50 | | Swin-B | | DeiT | | Avg. | |
| | AUROC↑ | FPR95↓ | AUROC↑ | FPR95↓ | AUROC↑ | FPR95↓ | AUROC↑ | FPR95↓ | AUROC↑ | FPR95↓ |
| MSP (Hendrycks & Gimpel, 2017) | 88.89 | 43.46 | 73.97 | 70.98 | 81.29 | 63.49 | 79.8 | 66.97 | 80.99 | 61.23 |
| Energy (Liu et al., 2020) | 94.11 | 27.56 | 79.53 | 65.8 | 80.07 | 60.35 | 71.65 | 72.65 | 81.34 | 56.59 |
| ReAct (Sun et al., 2021) | 94.07 | 27.69 | 83.34 | 54.81 | 85.2 | 53.53 | 77.16 | 68.74 | 84.94 | 51.19 |
| ODIN (Liang et al., 2018) | 93.73 | 29.68 | 79.42 | 65.77 | 80.68 | 58.94 | 76.07 | 66.43 | 82.47 | 55.2 |
| maxLogit (Basart et al., 2022) | 93.73 | 29.68 | 79.42 | 65.78 | 80.94 | 59.56 | 76.43 | 66.38 | 82.63 | 55.35 |
| Mahalanobis (Lee et al., 2018) | 94.27 | 27.11 | 68.36 | 80.63 | 87.86 | 52.07 | 83.98 | 73.86 | 83.62 | 58.42 |
| KLMatch (Basart et al., 2022) | 87.8 | 44.39 | 76.09 | 69.93 | 81.83 | 63.85 | 82.68 | 67.24 | 82.1 | 61.35 |
| KNN (Sun et al., 2022) | 92.6 | 34.38 | 84.43 | 57.46 | 85.08 | 65.24 | 82.75 | 76.24 | 86.22 | 58.33 |
| VIM (Wang et al., 2022) | 94.23 | 27.32 | 83.93 | 65.92 | 86.77 | **51.33** | 83.91 | 71.13 | 87.21 | 53.93 |
| FDBD (Liu & Qin, 2023) | 93.36 | 31.71 | 84.47 | 60.35 | 86.57 | 55.75 | 82.78 | 71.84 | 86.79 | 54.91 |
| Neco (Ammar et al., 2024) | 94.38 | 27.08 | 75.15 | 70.27 | 81.73 | 54.74 | 79.2 | **62.03** | 82.61 | 53.53 |
| WDiscOOD (Chen et al., 2023) | **94.41** | **26.35** | 70 | 78.71 | - | - | 83.97 | 73.83 | 82.8 | 59.63 |
| ours | 94.27 | **26.94** | **87.43** | **51.77** | **88.1** | **51.89** | **84.97** | 68.29 | **88.69** | **49.72** |

*Table 3.* **Comparison on DINO** Comparision with different post-hoc OOD detection methods on DINO. We report the averaged AUROC and FPR95 results over 6 OOD datasets. The detailed results can be viewed in the appendix.

| Method | DINO | |
|---|---|---|
| | AUROC↑ | FPR95↓ |
| SSD (Sehwag et al., 2021) | 51.5 | 96.86 |
| Neco (Ammar et al., 2024) | 46.97 | 96.69 |
| KNN (Sun et al., 2022) | 84.99 | 63.88 |
| Mahalanobis (Lee et al., 2018) | 91.57 | 39.27 |
| ours | **91.65** | **38.23** |

based on the cluster means $\{\mu_i\}$ with the minimum Mahalanobis distance to $f$. Figure 6 illustrates the value of $p$ and $q$ derived from ImageNet-1k benchmark experiments. As we can see from the figure, both $p$ and $q$ consistently exceed 1 by a considerable margin, even with ResNet-50 which produces less compact embedding space. This strongly supports our theoretical assumption. We also present the values of $s = f_r^\top \Sigma_R^{-1}(\mu_c - f_r')$ in the appendix on both ViT and ResNet-50.

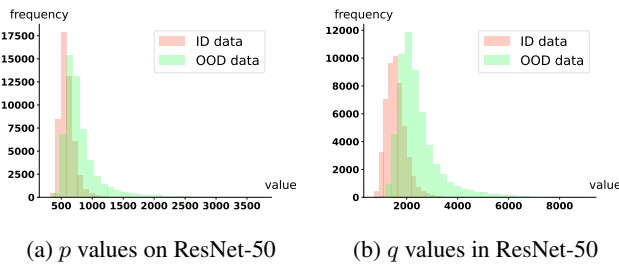

(a) $p$ values on ResNet-50     (b) $q$ values in ResNet-50

*Figure 6.* $p$ and $q$ values on ImageNet-1k pre-trained ResNet-50

### 5.8. Vector Norms in Residual Space

In addition to examining the effectiveness of projection in residual space based on performance metrics, we investigate further how residual projection helps OOD detection by analyzing feature scale. Given that all real-time features are normalized before conducting the prior adjustment, the norm of features before projected into the residual space is 1. But in residual space, the norm of $\mathbf{a}^\top \mathbf{B}$ can be smaller and with higher discriminative power. We visualize the distribution of norms in Figure 7. As shown in the figure, the OOD features tend to have larger norms in residual space for both ViT and ResNet-50 backbones. This distinction in norm distributions confirms that residual projection helps distinguish OOD samples from ID samples

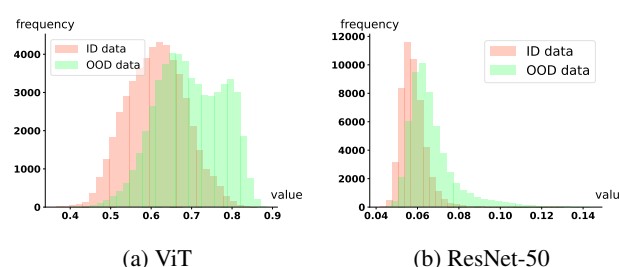

(a) ViT          (b) ResNet-50

*Figure 7.* The $l_2$ norms of features extracted by ImageNet-1k pre-trained ResNet-50 and ViT.

## 6. Conclusion

In this paper, we introduce a new perspective for defining distance-based OOD scores that dynamically refine the distorted geometry of the training distribution. Building on this perspective, we design an OOD detection score that adjusts the covariance in the Mahalanobis distance in real-time. Specifically, we restrict the adjustment to the residual space to avoid unintended influences. Extensive experiments demonstrate the effectiveness of our method. We also conduct experiments using the Euclidean distance and hard

*Table 4.* **Ablation study on different procedures.** The smaller value of the reported FPR95(%) means the better performance. **DME**, **RSP** and **DCM** are short for *Dynamic Matrix Estimation*, *Residual Space Projection* and *Dynamic Covariance Modeling* respectively.

| DME | RSP | DCM | CIFAR-10 DenseNet | | CIFAR-100 DenseNet | | ImageNet-1k ViT | | ImageNet-1k ResNet50 | |
|---|---|---|---|---|---|---|---|---|---|---|
| | | | AUROC↑ | FPR↓ | AUROC↑ | FPR↓ | AUROC↑ | FPR↓ | AUROC↑ | FPR↓ |
| | | | 95.19 | 18.86 | 89.47 | 35.84 | 92.51 | 33.16 | 82.49 | 60.43 |
| ✓ | | | 95.47 | 17.51 | 88.05 | 36.61 | 92.65 | 32.31 | 81.46 | 61.77 |
| ✓ | ✓ | | 94.08 | 23.92 | 85.69 | 53.70 | 82.72 | 86.32 | 87.31 | 52.17 |
| ✓ | | ✓ | 96.24 | 16.38 | 92.17 | 30.16 | 94.24 | 27.05 | 81.88 | 60.49 |
| ✓ | ✓ | ✓ | 96.83 | 14.63 | 92.38 | 29.98 | 94.27 | 26.94 | 87.43 | 51.77 |

OOD detection in the appendix material, illustrating that our method can be applied to various distance metrics. We plan to explore the design of $\mathcal{M}$ for vision language models and large language models in future work.

## Acknowledgements

This work is partially supported by Australian Research Council Project FT230100426.

## Impact Statement

This paper presents a methodological enhancement in out-of-distribution (OOD) detection, which is crucial for deploying AI systems in safety-critical environments. This addresses significant risks in sectors like autonomous transportation and medical diagnostics, where unrecognized OOD samples could lead to failures with dire consequences. Our approach, thus, not only advances the technical field of OOD detection but also contributes to societal trust in AI technologies by enhancing their reliability and safety.

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

## A. Theorical Proof

***Proof of Theorem 4.2.*** With Sherman-Morrison Formula, we have $d(f) = (f-a)\Sigma^{-1}(f-a) + \frac{((f-a)^\top \Sigma^{-1} f)^2}{1 - f\Sigma^{-1} f}$.

For $p = f^\top \Sigma^{-1} f < 1$, obviously $d(f) \geq 0$.

For $p = f^\top \Sigma^{-1} f > 1$, we have

$$
\begin{aligned}
d(f) =& p + q - 2s + \frac{(p-s)^2}{1-p} \geq 0 \\
\iff& p + q - pq - 2s + s^2 \leq 0 \\
\iff& s^2 - 2s \leq pq - p - q \\
\iff& (s-1)^2 \leq (p-1)(q-1).
\end{aligned}
$$

$\square$

## B. Additional Experiments

### B.1. Analyzing $s$ Values

In Section 4.3, we argue that $s = f_r^\top \Sigma_R(\mu_c - f_r')$ values can be positive for within-class covariance, where $\mu_c$ is selected based on the class means $\{\mu_i\}$ with the minimum Mahalanobis distance to real-time features $f$. Figure 8 shows the distribution of $s$ value on ImageNet-1k pre-trained ViT and ResNet-50. As shown in Figure 8, the $s$ scores are always positive, supporting our assumption that $\mathcal{M}$ induces a valid distance.

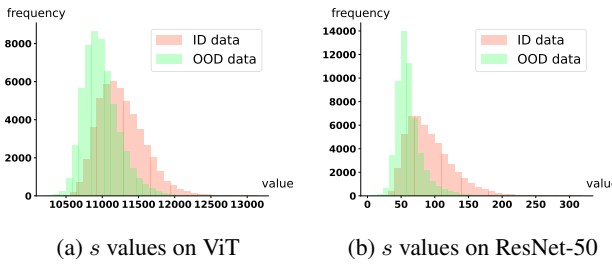

(a) $s$ values on ViT      (b) $s$ values on ResNet-50

*Figure 8.* The $s$ values on ImageNet-1k pre-trained ResNet-50 and ViT.

### B.2. Dynamic Adjustment in Euclidean Distance

Our approach can be utilized with various distances which utilize covariance matrix structures. For instance, if we consider $\Sigma$ as the identity matrix $I$, which corresponds to the Euclidean distance, then $\mathcal{M} = (I - f_r f_r^T)^{-1}$. We compare the OOD score with and without applying our approach in Euclidean distance in Table 5. All the features are normalized in the first stage. We conduct experiments on large-scale OOD detection tasks with 3 ImageNet pre-trained models, namely ViT, ResNet-50 and DINO, and CIFAR-10/100 pret-trained DenseNet. As shown in Table 5, better performance can be achieved with dynamic adjustment in

most cases. However, the improvement is insignificant since other factors, such as geometry in the principal space, affect the performance.

*Table 5.* **Ablation study on Euclidean distance.** We report the performance on ImageNet-1k pre-trained ViT, ResNet and DINO, and CIFAR-10/100 pre-trained DenseNet.

| Model | w adjustment | | w/o adjustment | |
|---|---|---|---|---|
| | AUROC↑ | FPR95↓ | AUROC↑ | FPR95↓ |
| CIFAR10 | | | | |
| DenseNet | 96.443 | 18.023 | 96.429 | 18.101 |
| CIFAR100 | | | | |
| DenseNet | 89.563 | 39.836 | 89.217 | 41.032 |
| ImageNet | | | | |
| ViT | 93.993 | 28.691 | 93.267 | 32.817 |
| ResNet-50 | 87.794 | 46.162 | 87.794 | 46.163 |
| DINO | 89 | 46.425 | 88.095 | 49.891 |

### B.3. Near OOD detection

We conduct near OOD detection on OpenOOD benchmark (Yang et al., 2022) with ResNet18. We report the experimental results in Table 6. We also compare with the results of the recent distance score and subspace score available in OpenOOD, namely KNN and VIM. All the results are averaged over 3 runs. We also provide the results of RMDS (Ren et al., 2021) with our covariance dynamic adjustment. RMDS is a Mahalanobis distance tailored for near-OOD detection.

As the table shows, vanilla Mahalanobis distance struggles to perform well on the near OOD detection task, which aligns with observations in previous work (Tajwar et al., 2021). In contrast, significantly enhances the effectiveness of the Mahalanobis distance in near OOD detection. In addition, it can also improve the performance of RMDS, reflecting the robustness of our method in different distances in near OOD detection.

## C. Algorithm Details

We present the pseudocode for our method in Algorithm 1. The calculation of $\mathbf{B}$ is similar to the $R$ in residual space method (Wang et al., 2022). We derive the basis matrix $\mathbf{B}$ from the eigenspace of within-class covariance $\Sigma_R$.

**Why using within-class covariance $\mu_R$?** We aim for the data to form a manifold, allowing us to easily determine its geometry. However, the features from training distribution form separate clusters for different classes in the penultimate layer. The whole data can not form a manifold that requires any local patch to be homomorphic to the Euclidean space. A simple translation in a direction like $\mu$ can not change the separated structure of the data distribution. However, if we define the deviation feature with the class mean $\mu_c$, the

*Table 6.* **Near OOD detection on OpenOOD benchmark**. We conduct experiments under four settings. When using CIFAR-10 as the seen dataset, we evaluate on CIFAR-100 and TinyImageNet as unseen datasets. Conversely, when CIFAR-100 serves as the seen dataset, CIFAR-10 and TinyImageNet are used as the unseen datasets. Best results are highlighted in bold.

| Method | CIFAR10-CIFAR100 | | CIFAR10-Tiny | | CIFAR100-CIFAR10 | | CIFAR100-Tiny | | avg | |
|---|---|---|---|---|---|---|---|---|---|---|
| | FPR↓ | AUROC↑ | FPR↓ | AUROC↑ | FPR↓ | AUROC↑ | FPR↓ | AUROC↑ | FPR↓ | AUROC↑ |
| VIM | 52.33 | 87.03 | 44.16 | 88.88 | 70.79 | 72.15 | 54.92 | 77.73 | 55.56 | 81.46 |
| KNN | 38.76 | **89.59** | 30.89 | **91.51** | 72.69 | 77.1 | 49.68 | **83.29** | 48.01 | 85.37 |
| Mahalanobis | 64.51 | 79.48 | 59.06 | 80.71 | 88.88 | 54.6 | 80.75 | 60.31 | 73.30 | 68.78 |
| RMDS | 49.76 | 88.26 | 37.63 | 90.29 | 62.90 | 77.69 | 49.55 | 82.60 | 49.96 | 84.71 |
| Mahalanobis+dynamic (ours) | 61.38 | 84.73 | 52.95 | 86.54 | 75.24 | 71.72 | 60.35 | 76.34 | 62.48 | 79.83 |
| RMDS+dynamic (ours) | **38.70** | 89.42 | **30.47** | 91.45 | **61.70** | **78.21** | **48.85** | 82.81 | **44.93** | **85.48** |

*Table 7.* Comparison of runtime and AUROC on CIFAR-10 pre-trained DenseNet.

| | Time (s) | AUROC(%) |
|---|---|---|
| Feature extraction only | 218.42 | N/A |
| GradNorm | 473.70 | 92.6 |
| Mahalanobis | 221.53 | 85.9 |
| ours | 235.59 | **96.93** |

cluster from different classes can be translated to centered as zero point, which is also the mean of the whole data. This allows the score to be calculated over the denser distribution as a manifold.

---

**Algorithm 1** OOD Score $s(\cdot)$ Calculation

---

**Input:** Feature vector $f$, basis matrix $\mathbf{B}$, within-class covariance matrix $\Sigma_R$, class means $\{\mu_i\}$
**Output:** Score function $s(\cdot)$
Normalize $f$ using the normalizer
  $\mathbf{a} \leftarrow$ `torch.einsum`$('i, bi \rightarrow b', f, \mathbf{B})$
  $adj \leftarrow \mathbf{B}^\top \mathbf{a}^\top \mathbf{a}\mathbf{B}$
  $dygeo \leftarrow$ `torch.linalg.inv`$(\Sigma_R - adj)$
  $d \leftarrow f - \{\mu_i\}$
  $dists \leftarrow$ `torch.einsum`$('bi, ij, bj \rightarrow b', d, dygeo, d)$
  $score \leftarrow -$`torch.min`$(\sqrt{dists})$
**return** $score$

---

## D. Detailed Results for CIFAR Benchmarks

Here, we present the detailed results for Table 1 of the main paper in Table 8 and 9. More details can be viewed in our source code in the supplementary.

## E. Detailed Results for ImageNet Benchmarks

We present the detailed results for Table 2 and 3 of the main paper, in Table 10. More details are in our source code in the supplementary.

## F. Discussion

**OOD detection in Vision-Language model (VLM) and Large language model (LLM)** Limited by the computing resource, it is impossible for us to evaluate our method on some large foundation models. In addition, the out-of-distribution is hard to define due to the large train set. Even if WiscOOD (Chen et al., 2023) evaluates CLIP on ImageNet benchmark, we doubt the validation of the experiments since no verification has been stated that there is no overlap between the image-text training set and the OOD datasets in the benchmark. However, according to our promising performance on different backbones, we believe our dynamical adjustment can be helpful in OOD detection of VLM and LLM.

**Inference cost** The computing complexity of our method is $O(n^3)$, where $n$ is the feature size. It takes around 0.002 seconds to calculate the score from each feature with 16GB V100 GPU. We provide the detection time cost comparison over 10,000 CIFAR-10 instances with DenseNet as the backbone network in Table 7. We note that our method has a comparable time cost to Mahalanobis distance method. This is because in post-hoc OOD detection, the dominant computational cost lies in the feature extraction, which is shared across most methods. In addition, we compare with the time cost of GradNorm (Huang et al., 2021) in the table, which shows the effectiveness of Mahanobis distance-based score functions. So, we believe the inference cost of our method is affordable in practice, especially with more advanced equipment like A100.

## G. Visualizaiton of Motivation

Our method is built on the assumption that the outlier features can undermine the effectiveness of the Mahalanobis distance. Figure 9 illustrates our motivation. As the figure shows, when outlier points shift away from the in-distribution (ID) center towards some out-of-distribution (OOD) points, the Mahalanobis distance can be smaller between ID and these OOD points. This will affect the performance of OOD detection.

*Table 8.* **Comparison on CIFAR pretrained DenseNet.** We compare different post-hoc OOD detection methods on CIFAR benchmarks. The base model is DenseNet.

| | OOD datasets | | | | | | | | | | | | | |
|---|---|---|---|---|---|---|---|---|---|---|---|---|---|---|
| Method | SVHN | | Texture | | LSUN_Resize | | LSUN | | iSUN | | Places365 | | AVG | |
| | AUROC↑ | FPR95↓ | AUROC↑ | FPR95↓ | AUROC↑ | FPR95↓ | AUROC↑ | FPR95↓ | AUROC↑ | FPR95↓ | AUROC↑ | FPR95↓ | AUROC↑ | FPR95↓ |
| | **CIFAR-10** | | | | | | | | | | | | | |
| MSP | 93.57 | 47.25 | 88.14 | 64.33 | 94.55 | 42.11 | 95.54 | 33.68 | 94.49 | 42.54 | 88.73 | 62.39 | 92.5 | 48.72 |
| Energy | 94.19 | 38.88 | 86.41 | 56.33 | 98.19 | 8.79 | 99.15 | 3.84 | 98.11 | 9.58 | 91.82 | 39.75 | 94.65 | 26.2 |
| maxLogit | 94.32 | 37.95 | 86.5 | 56.06 | 98.12 | 9.45 | 99.09 | 4.21 | 98.05 | 10.13 | 91.77 | 40.37 | 94.64 | 26.36 |
| ODIN | 94.33 | 37.83 | 86.52 | 56.05 | 98.12 | 9.48 | 99.09 | 4.21 | 98.05 | 10.15 | 91.77 | 40.37 | 94.65 | 26.35 |
| MahaVanilla | 98.11 | 7.93 | 92.81 | 25.66 | 87.91 | 51.71 | 82.62 | 67.99 | 88.79 | 45.52 | 65.15 | 87.02 | 85.9 | 47.64 |
| KNN | **99.29** | **3.96** | 96.39 | 19.61 | 98.12 | 9.9 | 98.75 | 6.91 | 98.2 | 10.26 | 89.95 | 46.33 | 96.79 | 16.16 |
| ReAct | 95.23 | 33.4 | 91.29 | 48 | 98.4 | 7.51 | 99.06 | 4.63 | 98.27 | 8.44 | 92.3 | 39.54 | 95.76 | 23.59 |
| Line | 97.75 | 11.42 | 95.11 | 23.44 | 99.1 | 4.12 | 99.83 | 0.62 | 99.02 | 5.08 | 91.13 | 43.85 | 96.99 | 14.75 |
| DICE | 94.96 | 27.82 | 86.96 | 46.03 | 99.06 | 4.23 | **99.9** | **0.38** | 98.99 | 5.2 | 90.18 | 45 | 95.01 | 21.44 |
| FDBD | 98.61 | 6.2 | 95.94 | 23.07 | 98.82 | 5.56 | 99.32 | 3.49 | 98.77 | 5.85 | **91.94** | **38.97** | **97.23** | **13.86** |
| ours | 98.41 | 8.21 | **96.41** | **16.51** | 99.34 | 3.01 | 99.82 | 0.71 | **99.17** | **4.04** | 87.84 | 55.28 | 96.83 | 14.63 |
| | **CIFAR-100** | | | | | | | | | | | | | |
| MSP | 75.18 | 82.01 | 71.41 | 84.8 | 69.18 | 85.22 | 85.6 | 60.5 | 70.17 | 86.01 | 74.71 | 82.24 | 74.37 | 80.13 |
| Energy | 81.3 | 88.02 | 71.01 | 84.33 | 80.14 | 70.69 | 97.43 | 14.75 | 78.95 | 74.6 | 78.15 | 78.25 | 81.17 | 68.44 |
| maxLogit | 81.42 | 86.26 | 71.18 | 83.44 | 79.77 | 71.7 | 97.06 | 16.98 | 78.69 | 75.05 | 78.22 | 77.78 | 81.06 | 68.53 |
| ODIN | 81.43 | 86.24 | 71.19 | 83.42 | 79.77 | 71.71 | 97.06 | 16.99 | 78.69 | 75.03 | 78.23 | 77.77 | 81.06 | 68.53 |
| MahaVanilla | 88.01 | 54.26 | 92.15 | 28.32 | 92.08 | 38.33 | 44.43 | 96.47 | 92.38 | 36.09 | 56.29 | 95.04 | 77.56 | 58.08 |
| KNN | **96.35** | **17.84** | 93.7 | 24.29 | 90.41 | 47.34 | 92.85 | 31.46 | 91.9 | 39.69 | 60.12 | 93.21 | 87.56 | 42.3 |
| ReAct | 82.77 | 83.69 | 77.7 | 79.79 | 81.71 | 71.56 | 97.13 | 15.82 | 81.06 | 75.19 | 77.53 | 78.25 | 82.98 | 67.38 |
| Line | 91.9 | 31.13 | 87.91 | 39.24 | 94.95 | 23.37 | 98.85 | 5.77 | 95.13 | 22.64 | 63.82 | 88.5 | 88.76 | 35.11 |
| DICE | 88.2 | 59.95 | 77.13 | 61.44 | 88.25 | 54.93 | **99.74** | **0.91** | 88.52 | 52.43 | 77.45 | 80.34 | 86.55 | 51.66 |
| FDBD | 91.38 | 49.75 | 91.59 | 43.3 | 88.71 | 57.75 | 96.54 | 19.19 | 89.43 | 56.03 | **77.87** | **77.39** | 89.25 | 50.57 |
| ours | 95.53 | 23.28 | **95.68** | **19.8** | 96.12 | 22.07 | 98.78 | 6.05 | **96.39** | **21.11** | 71.79 | 87.6 | **92.38** | **29.98** |

*Table 9.* **Comparison on CIFAR pretrained WideResNet.** We applied post-hoc OOD detection methods on CIFAR benchmarks. The base model is WideResNet.

| | OOD datasets | | | | | | | | | | | | | |
|---|---|---|---|---|---|---|---|---|---|---|---|---|---|---|
| Method | SVHN | | Texture | | LSUN_Resize | | LSUN | | iSUN | | Places365 | | AVG | |
| | AUROC↑ | FPR95↓ | AUROC↑ | FPR95↓ | AUROC↑ | FPR95↓ | AUROC↑ | FPR95↓ | AUROC↑ | FPR95↓ | AUROC↑ | FPR95↓ | AUROC↑ | FPR95↓ |
| | **CIFAR-10** | | | | | | | | | | | | | |
| MSP | 90.1 | 54.89 | 87.17 | 63.88 | 92.89 | 47.46 | 97.02 | 23.05 | 91.32 | 52.97 | 87.93 | 61.6 | 91.07 | 50.64 |
| Energy | 88.49 | 44.99 | 84.24 | 58.9 | 95.47 | 23.05 | 99.27 | 2.88 | 93.99 | 30.05 | 89.69 | 42.55 | 91.86 | 33.74 |
| MahaVanilla | 97.02 | 16.45 | 95.99 | 20.62 | 89.93 | 59.07 | 90.8 | 55.79 | 89.43 | 58.24 | 82.1 | 75.29 | 90.88 | 47.58 |
| KNN | 95.01 | 31.32 | 92.26 | 40.6 | 95.05 | 27.53 | 95.33 | 26.85 | 93.93 | 30.98 | **90.47** | **44.06** | 93.68 | 33.56 |
| maxLogit | 88.57 | 44.11 | 84.41 | 57.7 | 95.38 | 23.43 | 99.17 | 3.25 | 93.9 | 30.36 | 89.62 | 42.79 | 91.84 | 33.61 |
| ReAct | 88.63 | 45.4 | 86.24 | 57.45 | 95.05 | 24.43 | 99.23 | **3.21** | 93.52 | 31.62 | 89.86 | 42.23 | 92.09 | 34.06 |
| ODIN | 88.58 | 44.11 | 84.42 | 57.71 | 95.38 | 23.42 | 99.17 | 3.29 | 93.9 | 30.38 | 89.62 | 42.81 | 91.85 | 33.62 |
| GEM | 97.56 | 13.26 | **97.3** | **15.32** | 93.16 | 43.03 | 94.11 | 39.64 | 92.67 | 44.01 | 84.51 | 68.43 | 93.22 | 37.28 |
| DICE | 86.31 | 45.88 | 81.07 | 60.11 | 95.12 | 22.8 | 99.74 | 0.48 | 93.71 | 29.36 | 86.93 | 48.03 | 90.48 | 34.44 |
| Line | 71.92 | 73.62 | 70.2 | 76.37 | 82.27 | 61.36 | 96.78 | 12.11 | 79.04 | 69.76 | 73.46 | 76.36 | 78.94 | 61.6 |
| FDBD | 92.8 | 38.52 | 89.03 | 48.55 | 93.38 | 32.82 | 97.65 | 13.1 | 91.4 | 38.94 | 89.36 | 49.29 | 92.27 | 36.87 |
| ours | **97.73** | **11.31** | 96.95 | 16.44 | **97.08** | **15.86** | 99.29 | 3.46 | **96.63** | **18.89** | 89.38 | 45.44 | **96.18** | **18.57** |
| | **CIFAR-100** | | | | | | | | | | | | | |
| MSP | 70.91 | 85.24 | 71.39 | 85.89 | 79.2 | 79.82 | 88.56 | 49.1 | 78.1 | 81.57 | 73.41 | 83.26 | 76.93 | 77.48 |
| Energy | 70.87 | 87.59 | 72.85 | 85.27 | 82.62 | 78.49 | 96.83 | 16.74 | 80.89 | 82.51 | 74.94 | 81.08 | 79.83 | 71.95 |
| MahaVanilla | 89.68 | 44.91 | 90.03 | 42.38 | 91.73 | 38.28 | 47.63 | 99.71 | 91.16 | 40.56 | 65.86 | 91.94 | 79.35 | 59.63 |
| KNN | 89.95 | 43.03 | 89.74 | 40.99 | **94.32** | **29.49** | 82.02 | 56.3 | 92.32 | 34.47 | 69.68 | 85.65 | 86.34 | 48.32 |
| maxLogit | 71.22 | 86.59 | 73.07 | 85.07 | 82.78 | 78.18 | 96.19 | 21.5 | 81.14 | 81.89 | 75.13 | **80.99** | 79.92 | 72.37 |
| ReAct | 74.61 | 87.63 | 76.22 | 84.33 | 81.27 | 79.76 | **96.77** | **17.1** | 79.92 | 83.53 | **75.36** | 81.22 | 80.69 | 72.26 |
| ODIN | 71.23 | 86.58 | 73.07 | 85.09 | 82.78 | 78.2 | 96.19 | 21.51 | 81.15 | 81.93 | 75.13 | 80.99 | 79.93 | 72.38 |
| GEM | 90.42 | 43.98 | 90.85 | 39.69 | 93.69 | 33.7 | 60.29 | 98.22 | 92.31 | 36.98 | 68.68 | 90.34 | 82.71 | 57.15 |
| DICE | 68.38 | 90.41 | 72.4 | 83.07 | 78.8 | 81.68 | 98.68 | 4.18 | 78.2 | 83.64 | 74.16 | 83.27 | 78.44 | 71.04 |
| Line | 66.08 | 94.43 | 70.86 | 79.59 | 56.42 | 97.83 | 92.64 | 34.63 | 58.88 | 98.33 | 53.08 | 95.88 | 66.33 | 83.45 |
| FDBD | 83.18 | 73.01 | 83.26 | 71.97 | 89.75 | 58.07 | 91.16 | 45.85 | 88.16 | 63.22 | 75.31 | 82.51 | 85.14 | 65.77 |
| ours | **92.64** | **36.07** | **92.55** | **34.66** | 94.04 | 32.29 | 87.75 | 50.83 | **93.14** | **33.48** | 74.35 | 82 | **89.08** | **44.89** |

*Table 10.* **Comparison on ImageNet-1k pretrained models.** We applied post-hoc OOD detection methods on ImageNet benchmarks. The base models are Vit, ResNet-50, Swin-B, DeiT and DINO.

| Method | Texture | | SUN | | Places | | iNaturalist | | ImageNet-O | | OpenImage-O | | AVG | |
|---|---|---|---|---|---|---|---|---|---|---|---|---|---|---|
| | AUROC↑ | FPR95↓ | AUROC↑ | FPR95↓ | AUROC↑ | FPR95↓ | AUROC↑ | FPR95↓ | AUROC↑ | FPR95↓ | AUROC↑ | FPR95↓ | AUROC↑ | FPR95↓ |
| | | | | | | | **ViT** | | | | | | | |
| MSP | 85.42 | 52.43 | 86.93 | 53.22 | 85.72 | 57.75 | 96.97 | 13.63 | 85.81 | 51.75 | 92.48 | 31.99 | 88.89 | 43.46 |
| Energy | 91.25 | 36.13 | 93.28 | 34.44 | 90.98 | 42.82 | 98.94 | 5.6 | 93.36 | 30.3 | 96.87 | 16.07 | 94.11 | 27.56 |
| MahaVanilla | 91.71 | 36.21 | 92.82 | 35.35 | 89.51 | 46.22 | 99.77 | 1.08 | 94.28 | 30.1 | 97.51 | 13.7 | 94.27 | 27.11 |
| KNN | 90.81 | 38.39 | 90.46 | 46.67 | 87.16 | 54.8 | 98.66 | 6.81 | 92.47 | 39 | 96.06 | 20.6 | 92.6 | 34.38 |
| VIM | 91.47 | 37.75 | 93.3 | **32.64** | 89.77 | 44.44 | 99.66 | 1.55 | 94.09 | 31.05 | 97.11 | 16.48 | 94.23 | 27.32 |
| maxLogit | 90.86 | 38.56 | 92.81 | 37.45 | 90.66 | 44.65 | 98.81 | 6.03 | 92.69 | 33.55 | 96.54 | 17.83 | 93.73 | 29.68 |
| KLMatch | 84.9 | 51.86 | 85.11 | 56.62 | 83.56 | 61.49 | 96.36 | 13.7 | 85.16 | 50.9 | 91.7 | 31.76 | 87.8 | 44.39 |
| ReAct | 91.17 | 36.35 | 93.22 | 34.55 | 90.83 | 43.32 | 98.93 | 5.63 | 93.4 | 30.3 | 96.88 | 16.01 | 94.07 | 27.69 |
| ODIN | 90.86 | 38.56 | 92.81 | 37.46 | 90.66 | 44.66 | 98.81 | 6.03 | 92.69 | 33.55 | 96.54 | 17.83 | 93.73 | 29.68 |
| FDB | 90.54 | 39.17 | 92 | 40.7 | 89.59 | 48 | 98.76 | 6.59 | 92.77 | 36.6 | 96.49 | 19.23 | 93.36 | 31.71 |
| Neco | **91.92** | 35.37 | **93.79** | 32.85 | **91.12** | **42.39** | 99.01 | 5.15 | 93.42 | 30.8 | 97.01 | 15.9 | 94.38 | 27.08 |
| WDiscOOD | 91.83 | 35.87 | 93.19 | 33.32 | 89.77 | 44.35 | **99.79** | **1.03** | **94.41** | 29.55 | 97.49 | 13.95 | **94.41** | 26.35 |
| ours | 91.84 | **35.09** | 92.75 | 35.89 | 89.43 | 46.2 | 99.76 | 1.18 | 94.31 | 29.9 | **97.55** | 13.36 | 94.27 | 26.94 |
| | | | | | | | **ResNet-50** | | | | | | | |
| MSP | 80.46 | 66.13 | 81.75 | 68.58 | 80.63 | 71.57 | 88.42 | 52.77 | 28.64 | 100 | 83.91 | 66.84 | 73.97 | 70.98 |
| Energy | 86.73 | 52.29 | 86.73 | 58.28 | 84.13 | 65.4 | 90.59 | 53.95 | 41.92 | 100 | 87.06 | 64.88 | 79.53 | 65.8 |
| MahaVanilla | 89.98 | 43.48 | 52.02 | 97.37 | 51.68 | 97.33 | 63.9 | 93.78 | 80.94 | 66.35 | 71.63 | 85.46 | 68.36 | 80.63 |
| KNN | 97.49 | 10.85 | 80.71 | 69.64 | 74.86 | 78.03 | 86.02 | 59.55 | **84.62** | **62.4** | 82.89 | 64.29 | 84.43 | 57.46 |
| VIM | 96.86 | 14.79 | 81.1 | 82.05 | 78.42 | 83.27 | 87.53 | 71.33 | 70.87 | 85.05 | 88.81 | 59.01 | 83.93 | 65.92 |
| maxLogit | 86.4 | 54.33 | 86.59 | 59.9 | 84.18 | 65.68 | 91.13 | 50.87 | 40.86 | 100 | 87.36 | 63.9 | 79.42 | 65.78 |
| KLMatch | 82.82 | 64.77 | 80.55 | 73.51 | 78.86 | 75.56 | 89.91 | 44.29 | 38.67 | 100 | 85.74 | 61.48 | 76.09 | 69.93 |
| ReAct | 90.13 | 45.85 | **90.51** | **45.28** | **88.11** | **53.66** | 94.75 | 29.86 | 46.86 | 99.85 | 89.66 | 54.38 | 83.34 | 54.81 |
| ODIN | 86.4 | 54.31 | 86.59 | 59.9 | 84.18 | 65.67 | 91.14 | 50.86 | 40.86 | 100 | 87.36 | 63.88 | 79.42 | 65.77 |
| FDB | 92.1 | 37.52 | 86.81 | 61.25 | 84.07 | 67.11 | 93.71 | 40.16 | 59.83 | 100 | **90.3** | 56.07 | 84.47 | 60.35 |
| Neco | 95.62 | 17.8 | 66.46 | 87.25 | 66.44 | 88.28 | 69.17 | 83.39 | 75.88 | 73.3 | 77.32 | 71.61 | 75.15 | 70.27 |
| WDiscOOD | 91.62 | 38.83 | 54.64 | 96.64 | 53.65 | 96.82 | 65.98 | 91.98 | 82.16 | 64.1 | 71.95 | 83.9 | 70 | 78.71 |
| ours | **98.81** | **5.48** | 84 | 65.53 | 78.8 | 75.55 | 91.62 | 46.88 | 82.18 | 66.1 | 89.16 | **51.08** | **87.43** | **51.77** |
| | | | | | | | **Swin-B** | | | | | | | |
| MSP | 82.84 | 61.81 | 83.9 | 63.29 | 83.67 | 64.57 | 89.92 | 44 | 62.14 | 89.4 | 85.25 | 57.87 | 81.29 | 63.49 |
| Energy | 83.88 | 53.37 | 81.03 | 62.43 | 80.18 | 63.5 | 88.16 | 43.54 | 65.04 | 83.3 | 82.14 | 55.97 | 80.07 | 60.35 |
| Maha | 88.97 | 47.07 | 86.78 | 60.65 | 85.53 | 64.62 | 95.67 | 21.05 | 77.13 | 83.45 | 93.1 | 35.58 | 87.86 | 52.07 |
| KNN | 88.73 | 48.51 | 83.2 | 74.95 | 81.44 | 76.69 | 91.32 | 51.92 | 76.32 | 84.45 | 89.47 | 54.93 | 85.08 | 65.24 |
| VIM | 88.26 | 45.02 | 84.35 | 59.07 | 80.88 | 64.07 | 95.93 | 22.45 | 77.89 | 81.15 | 93.29 | 36.24 | 86.77 | 51.33 |
| maxLogit | 83.72 | 56.1 | 82.65 | 60.16 | 82.04 | 61.54 | 89.69 | 39.77 | 63.54 | 86.2 | 84 | 53.57 | 80.94 | 59.56 |
| KLMatch | 83.25 | 60.28 | 83.49 | 69.05 | 82.61 | 71.28 | 90.44 | 42.61 | 64.64 | 84.65 | 86.54 | 55.22 | 81.83 | 63.85 |
| ReAct | 87.33 | 50.83 | **86.89** | 54.38 | 85.87 | **57.2** | 93.31 | 31.27 | 68.67 | 82.8 | 89.13 | 44.7 | 85.2 | 53.53 |
| ODIN | 83.46 | 55.66 | 82.48 | 59.32 | 81.83 | 60.9 | 89.72 | 38.65 | 62.97 | 85.95 | 83.61 | 53.14 | 80.68 | 58.94 |
| FDB | 87.44 | 50.69 | 86.84 | 59.78 | **85.89** | 62.58 | 93.71 | 33.61 | 74.27 | 84.85 | 91.26 | 43.01 | 86.57 | 55.75 |
| neco | 82.65 | 54.5 | 84.17 | **53.65** | 83.35 | 56.32 | 93.82 | 24.96 | 60.66 | 92.65 | 85.73 | 46.36 | 81.73 | 54.74 |
| ours | **90.13** | **41.58** | 85.14 | 71.38 | 83.88 | 74.77 | **96.76** | **13.83** | **78.87** | **78.4** | **93.82** | 31.36 | **88.1** | **51.89** |
| | | | | | | | **DeiT** | | | | | | | |
| MSP | 81.59 | 64.56 | 80.72 | 68.35 | 80.37 | 70.04 | 88.46 | 51.18 | 63.24 | 87 | 84.39 | 60.71 | 79.8 | 66.97 |
| Energy | 77.5 | 65.37 | 70.43 | 75.96 | 69.17 | 77.48 | 78.02 | 67.14 | 60.12 | 83.3 | 74.65 | 66.63 | 71.65 | 72.65 |
| MahaVanilla | 82.91 | 78.33 | 81.93 | 78.53 | 80.88 | 77.29 | 92.14 | 55.62 | 76.26 | 90.55 | 89.73 | 62.82 | 83.98 | 73.86 |
| KNN | 86.57 | 59.91 | 78.98 | 84.68 | 76.97 | 84.1 | 88.64 | 73.98 | 77.56 | 87.1 | 87.8 | 67.7 | 82.75 | 76.24 |
| VIM | 83.3 | 75.96 | 81.93 | 74.79 | 80.06 | 74.2 | 92.9 | 49.94 | 75.43 | 89.7 | 89.84 | 62.22 | 83.91 | 71.13 |
| maxLogit | 80.26 | 61.38 | 76.12 | 68.45 | 75.44 | 70.18 | 85.23 | 53.28 | 61.04 | 84.7 | 80.52 | 60.31 | 76.43 | 66.38 |
| KLMatch | 84.14 | 63.37 | **82.34** | 72.91 | **81.56** | 74.76 | 90.8 | 49.64 | 69.7 | 83.5 | 87.56 | 59.26 | 82.68 | 67.24 |
| ReAct | 80.42 | 64.18 | 77.05 | 70.45 | 75.86 | 72.26 | 84.34 | 61.11 | 64.41 | 82.35 | 80.86 | 62.09 | 77.16 | 68.74 |
| ODIN | 80.01 | 61.47 | 75.63 | 68.62 | 74.94 | 70.47 | 85.13 | 52.92 | 60.54 | 84.6 | 80.15 | 60.47 | 76.07 | 66.43 |
| FDB | 82.66 | 70.09 | 81.18 | 74.8 | 80 | 74.58 | 89.47 | 62.74 | 75.16 | 86.5 | 88.22 | 62.32 | 82.78 | 71.84 |
| neco | 79.65 | 60.92 | 77.48 | **65.62** | 76.79 | **68.02** | 91.59 | 38.74 | 63.72 | 85 | 85.96 | **53.85** | 79.2 | **62.03** |
| WDiscOOD | 82.91 | 78.24 | 81.93 | 78.5 | 80.87 | 77.27 | 92.14 | 55.59 | 76.25 | 90.6 | 89.74 | 62.78 | 83.97 | 73.83 |
| ours | **87.15** | **58.24** | 79.68 | 89.49 | 78.25 | 89.63 | **94.43** | **37.42** | **78.85** | **80.55** | **91.45** | 54.41 | **84.97** | 68.29 |
| | | | | | | | **DINO** | | | | | | | |
| SSD | 50.61 | 97.34 | 52.6 | 96.71 | 51.83 | 96.08 | 53.84 | 99.59 | 47.27 | 95.15 | 52.82 | 96.27 | 51.5 | 96.86 |
| Neco | 50.65 | 93.85 | 45.33 | 98.02 | 46.98 | 98.06 | 39.77 | 98.44 | 51.56 | 95.25 | 47.51 | 96.52 | 46.97 | 96.69 |
| KNN | 93.8 | 26.51 | 83.82 | 73.04 | 79.63 | 76.29 | 87.76 | 65.14 | 81.32 | 75.45 | 83.6 | 66.84 | 84.99 | 63.88 |
| MahaVanilla | **96.84** | **13.35** | 90.62 | 46.95 | 87.48 | 55.01 | 96.99 | 14.53 | **84.77** | **67.3** | 92.71 | 38.49 | 91.57 | 39.27 |
| ours | 96.59 | 14.02 | **91.11** | **43.71** | **87.99** | **52.13** | **97.22** | **12.69** | 84.22 | 69.3 | **92.79** | **37.55** | **91.65** | **38.23** |

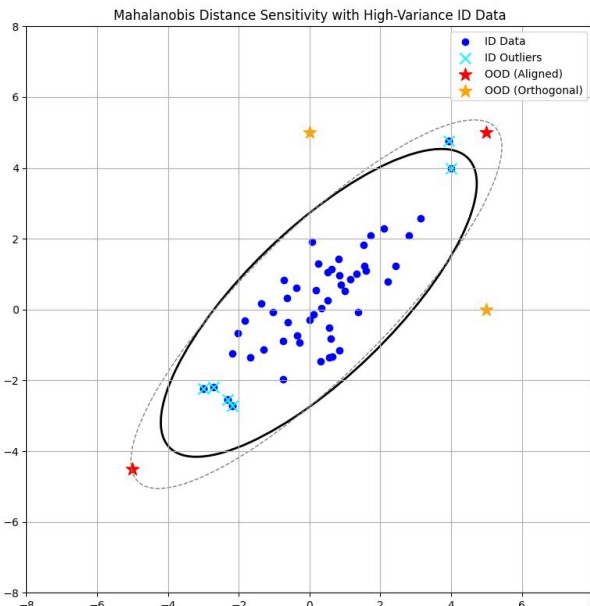

*Figure 9.* Visualization of the phenomenon that Mahalanobis distance may become less sensitive to OOD samples that align with the directions of high variance in ID data caused by outliers. The bold (solid) line is the geodesic line without manually added outlier points, and the dashed line is the geodesic line with the added outlier points. The two geodesic lines are on the same level.

