# OpenReview forum: "Improving Out-of-Distribution Detection via Dynamic Covariance Calibration"
_ICML.cc/2025/Conference — ICML 2025 poster_

### Official Review · Reviewer_6R2e · 2025-03-12

**Overall Recommendation:** 3

**Summary:**

To reduce redundant information in the covariance matrix in real-time OOD detection, this work proposes adjusting prior geometry based on the input, enhancing sensitivity to OOD samples while preserving essential information for ID classification.

**Claims And Evidence:**

Most claims are supported by clear and convincing evidence.

**Essential References Not Discussed:**

None

**Experimental Designs Or Analyses:**

The final results show that their method effectively improves OOD detection. However, the experiment in Section 5.8 does not convincingly demonstrate that their method increases the discrepancy in feature norms between ID and OOD samples.

**Methods And Evaluation Criteria:**

I think their method is reasonable, and the evaluation criteria used for OOD detection are appropriate.

**Other Comments Or Suggestions:**

1. STA in Line 67.

**Other Strengths And Weaknesses:**

Strengths:
1. This work proposes adapting the prior geometry based on the input, improving sensitivity to OOD samples while retaining essential information for ID classification.
2. Their idea is reasonable, and their implementation is novelty.
3. Experimental results show that their method outperforms SOTA.

Weakness.
1. Why the baseline methods provided in CIFAR benchmark and ImageNet benchmark.
2. The authors give evidence in 5.8 that their projection based on normalized features results in a larger norm for OOD and smaller norm for ID. However, this phenomenon has been widely observed [Park et al.] Can we make sure this discrepancy gets improved with projection, compared with feature norm without normalization.

Park, Jaewoo, et al. "Understanding the feature norm for out-of-distribution detection." Proceedings of the IEEE/CVF international conference on computer vision. 2023.

**Questions For Authors:**

None

**Relation To Broader Scientific Literature:**

Since Mahalanobis distance [Lee et al.] demonstrated the effectiveness of incorporating information geometry in designing distance-based scores, more studies have attempted to capture more precise geometry matrices $M$.  Wang and Ammar integrate information from residual space and principal space to achieve this but remain limited by static geometry. This work introduces input based projection matrix to address this limitation.

Lee, Kimin, et al. "A simple unified framework for detecting out-of-distribution samples and adversarial attacks." Advances in neural information processing systems 31 (2018).

Wang, Haoqi, et al. "Vim: Out-of-distribution with virtual-logit matching." Proceedings of the IEEE/CVF conference on computer vision and pattern recognition. 2022.

Ammar, Mouïn Ben, et al. "Neco: Neural collapse based out-of-distribution detection." arXiv preprint arXiv:2310.06823 (2023).

**Theoretical Claims:**

I did not notice any clear errors in their theoretical claims.

---

> ### Author Rebuttal · Authors · 2025-04-01
>
> We appreciate the reviewer's positive recognition of our work, particularly regarding the novelty and reasonableness of the idea, strong performance, and clarity of supporting evidence. Below, we provide detailed, point-by-point responses addressing each comment.
> ### Other Strengths And Weaknesses:
> **C1**: Why the baseline methods provided in CIFAR benchmark and ImageNet benchmark.
>
> **C1-Ans**:
> The reason that the compared methods are different in CIFAR and ImageNet benchmarks is that we can not access the hyperparameter settings in the absent methods. To address the author's concern, we reimplement and do hyperparameter search on VIM and NECO in CIFAR benchmark. In addition, we add a gradient-based method in this setting, namely GradeNorm [2].
>
> |           | CIFAR10 DenseNet (AUROC/FPR) | CIFAR100 DenseNet (AUROC/FPR) |
> |-----------|------------------------------|-------------------------------|
> | GradNorm  | 92.60/24.83                  | 79.75/63.08                   |
> | VIM       | 93.44/33.66                  | 87.95/51.39                   |
> | NECO| 94.51/27.92                  | 79.79/72.43                   |
> | ours      | 96.83/14.63                  | 92.38/29.98                   |
>
> **C2**: Can we make sure this discrepancy gets improved with projection, compared with feature norm without normalization.
>
> **C2-Ans**: Thank you for the insightful question. We have added the discussion about the observation of NAN[1] in our experiment. In the following table, we show the results w and w/o projection to illustrate the effectiveness of the projection.
> |                | ImageNet-ViT(AUROC/FPR) |ImageNet-ResNet50(AUROC/FPR)|
> |-------------------|---------------|----------------|
> | w/o projection| 94.24/27.05        |81.88/60.49|
> | w projection  | 94.27/26.94      |87.43/51.77|
>
> In addition, we use the norm of the projected feature as the OOD score to answer this question, shown in the following table. The results in the table imply that normalization can help improve the discrepancy. We present our explanation of this phenomenon in the following
>
> |  |ImageNet-ViT(AUROC/FPR) |ImageNet-ResNet50(AUROC/FPR)|
> |-------------------|---------------|----------------|
> | w/o normalizaiton| 92.34/34.80        |62.27/81.95|
> | w normalizaiton  | 92.80/33.40        |83.26/59.74|
>
>
> The $L_2$ norm of the feature can also be interpreted as the distance from the feature to the zero point. If the features in different semantic distributions have significantly different scales and the center of ID features is far from the zero point, the norm of the features makes it hard to detect the OOD sample very well, since the zero point may also be an OOD point. However, the normalized features are distributed on the sphere with radius 1 and the zero point as center. As long as the class clusters are not distributed too close, the center of normalized features can be easily near the zero points, thus making the norm of the features have geometrical meaning for OOD detection. If the center of the features is the zero point, the center of the projected features is also the zero point. We assume this is the reason that the normalization can help improve the discrepancy and also strengthen the neural collapse phenomenon
>
> [1] Park, Jaewoo, et al. "Understanding the feature norm for out-of-distribution detection." Proceedings of the IEEE/CVF international conference on computer vision. 2023.
> [2] Huang, Rui, et al."On the importance of gradients for detecting distributional shifts in the wild.

---

> > ### Comment · Reviewer_6R2e · 2025-04-02
> >
> > The experimental results of ImageNet-Vit w and w/o projection do not show much difference.

---

> > > ### Author Response · Authors · 2025-04-03
> > >
> > > Thank you for the constructive feedback. While the improvement on ImageNet–ViT is indeed modest, it becomes more pronounced on ResNet‐50 (AUROC: 81.88→87.43, FPR: 60.49→51.77). We hypothesize that ViT’s architecture naturally produces a well-conditioned embedding space, thus leaving less room for projection-based refinements. Additionally, prior work [1] suggests that the most distinct components for Mahalanobis scores often lie in the principal space. If the projected (residual) adjustment is relatively small compared to that principal-space effect, gains on already well-conditioned representations (e.g., ViT) may appear modest. Observations from our second table (with/without normalization) further indicate that ViT’s feature distribution is excellent, and the gap between the residual space and principal space can be large- its richest information is already captured in the principal subspace. We demonstrate this phenomenon with the smallest eigenvalue of the residual space and the non-residual space in the following. Thus, there is little room for additional improvement via our projection on ViT. Nevertheless, the gains, though smaller, remain consistent, demonstrating the broad applicability of our approach.
> > >
> > > ||ViT(residual/non-residual)|ResNet-50(residual/non-residual)|
> > > |-|-|-|
> > > |smallest eigen values|6.4$e^{-9}$/1.2$e^{-3}$|7.8$e^{-7}$/8.2$e^{-6}$|
> > >
> > > [1] Ren, Jie, et al. A simple fix to Mahalanobis distance for improving near-ood detection.

---

### Official Review · Reviewer_YqZR · 2025-03-13

**Overall Recommendation:** 3

**Summary:**

This paper addresses the problem of Out-of-Distribution (OOD) detection, which is critical for ensuring the reliability of AI systems.
The authors observe that while existing subspace-based methods use information geometry to detect OOD data, they fail to address the distortion in geometry caused by ill-distributed samples that can arise in training data.
To mitigate this issue, the paper proposes a novel approach that dynamically updates the prior covariance matrix using real-time input features.  This update reduces the covariance along the direction of real-time input features and constrains adjustments to the residual space. This preserves essential data characteristics and avoids unintended effects on other directions.
The method is evaluated on CIFAR and ImageNet datasets, demonstrating improved OOD detection across various models.

**Claims And Evidence:**

The central claim that dynamically adjusting the prior covariance matrix improves Out-of-Distribution (OOD) detection is supported by the experimental results. The authors consistently show improved performance on CIFAR and ImageNet datasets across various models when compared to mahalanobis distance specifically. The ablation studies further reinforce the importance of each component of their method (dynamic adjustment, residual space projection, and deviation features).

**Essential References Not Discussed:**

The paper effectively discusses relevant prior work in OOD detection, covering distance-based, subspace-based, and other categories. However, one area where additional discussion could provide valuable context is the limitations and potential failure modes of Mahalanobis distance. For example, [1] discusses that MSP baseline outperforms Mahalanobis when ID and OOD distributions are very similar can the adjustment help with that problem.

References:
* [1] Ren, Jie, et al. "A simple fix to mahalanobis distance for improving near-ood detection." arXiv preprint arXiv:2106.09022 (2021).
* [2] Tajwar, Fahim, et al. "No true state-of-the-art? ood detection methods are inconsistent across datasets." arXiv preprint arXiv:2109.05554 (2021).

**Experimental Designs Or Analyses:**

**1. Experimental Designs**

* **OOD Detection Benchmarks:** The authors use two standard OOD detection benchmarks: CIFAR (CIFAR-10/CIFAR-100) and ImageNet-1k. These are widely used datasets in the OOD detection community, making the evaluation relevant and comparable to other work. They include a variety of OOD datasets to test the generalization of their method.

* **Comparison with State-of-the-Art:** The authors compare their method with a diverse set of post-hoc OOD detection approaches, including probability-based, logit-based, density-based, distance-based, and subspace-based methods. This comprehensive comparison provides a strong baseline for evaluating the effectiveness of their proposed approach. The only family of methods that's missing is gradient based methods (for eg. GradNorm[1], GradOrth[2], GROOD[3])

* **Ablation Study:** The authors conduct a thorough ablation study to analyze the contribution of different components of their method. They systematically remove key components (residual space projection, real-time adjustment, and deviation features) and evaluate the impact on performance. This helps to demonstrate the importance of each component.

* **Analysis of Residual Dimensionality:** The authors analyze how the performance of their method varies with different dimensions of the residual space. This analysis provides insights into the sensitivity of the method to the choice of residual space dimensionality.

* **Analysis of p and q Values:** The authors analyze the values of p and q, which are important parameters in their proposed method. This analysis provides empirical support for the theoretical claims made in the paper.

* **Euclidean Distance Experiments:** The authors conduct additional experiments using Euclidean distance to demonstrate the general applicability of their approach to different distance metrics.

**2. Soundness and Validity**

* The experimental designs appear to be generally sound and well-justified. The authors use appropriate datasets, evaluation metrics, and comparison methods.

* The ablation studies are particularly strong, providing clear evidence for the contribution of each component of the proposed method.

* The analyses of residual dimensionality, p and q values, and vector norms provide valuable insights into the behavior of the method and support the theoretical claims.

* The inclusion of Euclidean distance experiments and hard OOD detection further demonstrates the robustness and generalizability of the approach.

References:
1- Huang, Rui, Andrew Geng, and Yixuan Li. "On the importance of gradients for detecting distributional shifts in the wild." Advances in Neural Information Processing Systems 34 (2021): 677-689.
2- Behpour, Sima, et al. "Gradorth: A simple yet efficient out-of-distribution detection with orthogonal projection of gradients." Advances in Neural Information Processing Systems 36 (2023): 38206-38230.
3- ElAraby, Mostafa, et al. "GROOD: GRadient-aware Out-Of-Distribution detection in interpolated manifolds." arXiv preprint arXiv:2312.14427 (2023).

**Methods And Evaluation Criteria:**

The proposed method is reasonable for the problem of OOD detection. The use of covariance matrix adaptation to capture and adjust information geometry is a sound approach. The evaluation criteria (AUROC and FPR95) and benchmark datasets (CIFAR and ImageNet) are standard and appropriate for assessing OOD detection performance


 **Experimental Design and Analysis:** The experimental design is comprehensive. The authors compare their method with a wide range of existing OOD detection techniques, providing a strong baseline for evaluation. They also perform ablation studies and analyze the behavior of key variables (p, q, and vector norms). The analysis is generally sound and provides strong support for their claims.

**Other Comments Or Suggestions:**

* **Line 110**: I guess we have type in select import neurons should be select improtant neurons

**Other Strengths And Weaknesses:**

**Strengths:**

* **Originality:** The paper introduces a novel perspective to distance-based OOD detection by dynamically adjusting the prior geometry using real-time input features. This approach is a departure from traditional methods that rely on static estimations of the information geometry from the training distribution. The idea of refining the covariance matrix in real-time and constraining adjustments to the residual space to preserve essential ID characteristics is an original contribution.

* **Significance:** OOD detection is a crucial problem for the trustworthiness and reliability of AI systems, especially in safety-critical applications. The paper addresses a significant limitation of existing methods and proposes a solution that improves OOD detection performance across various models and datasets. This has the potential to contribute to safer and more reliable AI deployments.

* **Clarity:** The paper is generally well-written and easy to follow.

**Weaknesses:**

* **Computational Complexity:** The paper mentions that the computational complexity of their method is O(n^3), where n is the feature size. While the authors argue that the inference cost is affordable, a more detailed discussion of the computational trade-offs compared to other methods would be beneficial.

* **Evaluation against Near-OOD** The test pairs used for testing against other OOD methods are mostly from Far-OOD, it would be beneficial to see how it performs against near-OOD data as per OpenOOD benchmark.

**Questions For Authors:**

- The paper mentions that the computational complexity of the method is O(n^3) (line 700). While the authors state that the inference cost is affordable, could they provide a more detailed comparison of the computational cost with other OOD detection methods?

- The paper argues that Mahalanobis distance may become less sensitive to OOD samples that align with the directions of high variance in ID data caused by outliers. Could the authors provide more specific examples or visualizations to illustrate this phenomenon? A clearer illustration would strengthen the motivation for the proposed method.

**Relation To Broader Scientific Literature:**

The key contribution of this paper is a novel approach to Out-of-Distribution (OOD) detection that dynamically adjusts the prior matrix using real-time input features. This contrasts with prior work that relies on static estimations of geometry from ID data.

Specifically, the paper builds upon distance-based OOD detection methods, including those using Mahalanobis distance, which utilize a covariance matrix derived from ID data to capture information geometry.. The authors identify a limitation in these methods: they often neglect the distortion of information geometry caused by outlier features in the training data.

The paper also relates to subspace-based methods that project features onto the subspace of ID data. While these can be seen as using matrix-induced distance scores, the authors argue that simply replacing the covariance matrix may remove important ID information and doesn't provide targeted adjustment for OOD directions. Their dynamic adjustment approach refines the prior information geometry using local information from real-time features. Additionally, they constrain adjustments to the residual space of the training distribution, building on the idea that OOD features may exhibit more energy in this space.

**Theoretical Claims:**

The primary theoretical claim in the paper is presented in Theorem 4.2.  This theorem provides the conditions under which the proposed dynamic distance metric is valid.

Theorem 4.2: Given a feature f, a non-zero feature a, and a symmetric positive definite matrix Σ, the theorem defines variables p, q, and s based on these inputs and Σ.  It then states that under certain conditions involving p, q, and s, the proposed distance metric d(f) is greater than or equal to 0.

The proof appears to be mathematically sound and follows a logical progression.

---

> ### Author Rebuttal · Authors · 2025-04-01
>
> We appreciate the reviewer's recognition of our work's **novelty**, **improved performance** demonstrated through comprehensive experiments, clear **presentation**, and its **significance** for AI system reliability. Below, we address each comment in detail.
>
> **C1**: gradient-based methods
>
> **C1-Ans**: Since the code of GradNorm[3] is the only implementation available, we compare our method with this gradient-based method. We have added these three works in Section 2
>
> Due to its computational cost, we only conduct the comparison on CIFAR benchmark. In the following, we present the results of GradNorm.
>
> || CIFAR10 (AUROC/FPR) | CIFAR100 (AUROC/FPR) |
> |-|-|-|
> |GradNorm|92.60/24.83|79.75/63.08|
> |ours|96.83/14.63|92.38/29.98|
>
> **C2**: additional discussion on limitations and potential failure modes of Mahalanobis distance
>
> **C2-Ans**: Previous work has discussed the failure within the dimension corresponding to the smaller eigenvalues of ID covariance [1] and less effective performance when ID and OOD are similar [2]. The strong performance of our method aligns with and supports the findings in [1]. Our approach explicitly adjusts the covariance matrix within the residual space, thereby mitigating the aforementioned vulnerability regarding the discrepancy of Mahalanobis distance in residual space. In addition, our adjustment can also help improve the performance of  Mahalanobis distance in near OOD detection. Details can be viewed in the answer of **C4**. We have added this discussion in our paper.
>
> **C3(Q1)**: Detailed discussion and comparison on computational complexity
>
> **C3(Q1)-Ans**: In the following table, we provide the detection time cost comparison over 10,000 CIFAR10 instances with DenseNet as the backbone network.
>
> ||Time(s)|CIFAR10(AUROC)|
> |-|-|-|
> |Feature extraction only|218.42|N/A|
> |Mahala|221.53|85.9|
> |ours|235.59|**96.93**|
> |GradNorm|473.7|92.6|
>
> We note that our method has a comparable time cost to Mahalanobis distance method. This is because in post-hoc OOD detection, the dominant computational cost lies in the feature extraction, which is shared across most methods. The time complexity we provided is only for the feature-level operation, which is insignificant as $n$ (feature dimension) is small.  In addition, we compare with the time cost of GradNorm in the table
>
> **C4**: Near-OOD on OpenOOD benchmark
>
> **C4-Ans**:  We conduct near OOD detection on OpenOOD benchmark with ResNet18. We report the experimental results in the following table. We also compare with the results of the recent distance score and subspace score available in OpenOOD, namely KNN and VIM. All the results are averaged over 3 runs. We also provide the results of RMDS [1] with our method. RMDS is a Mahalanobis distance tailored for near-OOD detection. As the table shows, our method can effectively improve the performance of Mahalanobis distance in near OOD detection. It can also improve the RMDS, reflecting the robustness of our method in different distances in near OOD detection.
> ||CIFAR10-CIFAR100||CIFAR10-Tiny||CIFAR100-CIFAR10|| CIFAR100-Tiny||avg||
> |-|-|-|-|-|-|-|-|-|-|-|
> ||FPR|AUROC|FPR|AUROC|FPR|AUROC|FPR|AUROC|FPR|AUROC
> |VIM|52.33|87.03|44.16|88.88|70.79|72.15|54.92|77.73|55.56|81.46|
> |KNN|38.76|**89.59**|30.89|**91.51**|72.69|77.1|49.68 | **83.29**|48.01|85.37|
> |mahala|64.51|79.48|59.06|80.71|88.88|54.6|80.75| 60.31|73.30|68.78|
> |mahala+dynamic(ours)|61.38|84.73|52.95|86.54|75.24|71.72|60.35|76.34|62.48|79.83|
> |RMDS| 49.76| 88.26| 37.63| 90.29| 62.90| 77.69| 49.55| 82.60|49.96|84.71|
> |RMDS+dynamic(ours)|**38.70**|89.42|**30.47**|91.45|**61.7**|**78.21**|**48.85**|82.81|**44.93**|**85.48**|
>
> **C5**: typos
>
> **C5-Ans**: Thank you for pointing out this typo. We have revised it in our paper.
> ### Questions
> **C6(Q2)**: Visualization of the phenomenon that Mahalanobis distance may become less sensitive to OOD samples that align with the directions of high variance in ID data caused by outliers
>
> **C6(Q2)-Ans**:  **Visualiztion**: https://i.imgur.com/U4rrTTt.png We visualise this phenomenon in 2D space. Note that, as the features are in the spaces with higher dimensions, the situation may be more complicated than our demonstration. Mahalanobis distance can be considered as a geodesic distance of normal distribution, thus we utilize the geodesic line within the randomly sampled normal distribution (ID data) to demonstrate this phenomenon. As the figure shows, the geodesic line may be made further from the center point by the outlier points.  It is obvious that the OOD points, aligned with outlier points, can be closer to the dashed line than the bold line, which makes the distance between OOD points and ID center smaller.
>
> [1] Ren, Jie, et al. A simple fix to Mahalanobis distance for improving near-ood detection.
> [2] Tajwar, Fahim, et al. No true state-of-the-art? OOD detection methods are inconsistent across datasets.
> [3] Huang, Rui, et al."On the importance of gradients for detecting distributional shifts in the wild.

---

### Official Review · Reviewer_WNVA · 2025-03-13

**Overall Recommendation:** 3

**Summary:**

This paper proposes a dynamic covariance calibration approach for OOD detection, addressing the sensitivity of distance-based detectors to outliers in the ID data. While existing methods mitigate this issue by projecting features onto principal dimensions, they risk losing valuable ID information. Instead, the proposed method dynamically adjusts the covariance matrix using real-time input features while preserving the ID structure by only updating the residual subspace.

## Update after rebuttal

Most of my concerns have been answered, especially regarding the novelty of the method, which differs from other subspace approaches due to test time adaptation of the OOD criterion.

**Claims And Evidence:**

The essential claim of the paper is that OOD detection might be improved by only considering In-Distribution (ID) subspaces. This claim is shared with other recent papers in the literature. Here, however, the ablation shows only FPR95 results but this metric is prone to large variations. AUC is more challenging and could better indicate the impact of each component of the method. Furthermore, the overall table is difficult to read as it does not show the full combinatorial related to each component.

**Essential References Not Discussed:**

Essential works are cited.

**Experimental Designs Or Analyses:**

t is essential that the proposed method is compared with other subspace-based OOD detection methods namely ViM and Neco also on the CIFAR benchmark. Same goes for Tab 3.

**Methods And Evaluation Criteria:**

The benchmarks are only considering far-OOD samples and should be supplemented with near-OOD samples. In particular, CIFAR benchmark could used Tiny-Imagenet as OOD and also add the experiments: C-10 (ID) vs C-100 (OOD) and C-100 (ID) vs C-10 (OOD). For the second benchmark, results on iNaturalist could bring more insights on the method performances for near-OOD detection. For fairer comparison and broader, it would have been interesting to have SOTA results on standard benchmarks such as OpenOOD 1.5.

**Other Comments Or Suggestions:**

* l. 266-268: class is indexed by $i$ then by $c$
* $s(z)$ does not depends on $z$ in the right side of eq. (3)

**Other Strengths And Weaknesses:**

The name of related work sub-section 2.2 ``Test time OOD detection'' is misleading as it might refer to test-time adaptation. Here the authors are more concerned about post-hoc methods.

The dynamic aspect of the method is difficult to grasp. As $\bm \Sigma_R$ and $\mathcal M(f)$ are computed on the training dataset, the parameters of the OOD criterion are fixed for any given test-sample.

**Questions For Authors:**

No questions

**Relation To Broader Scientific Literature:**

The proposed approach is particularly similar to ViM, which also operates on the residual subspace of the antepenultimate layer. The essential difference is the OOD score. Moreover, it shares also common ideas with necko. Differences and in particular the relevance of the proposed approach compared to these methods could be discussed in more detail.

ViM: Out-Of-Distribution with Virtual-logit Matching, Wang, Haoqi and Li, Zhizhong and Feng, Litong and Zhang, Wayne, CVPR 2022

NECO: NEural Collapse Based Out-of-distribution detection, Mouïn Ben Ammar, Nacim Belkhir, Sebastian Popescu, Antoine Manzanera, Gianni Franchi, ICLR 2024

**Theoretical Claims:**

I did not find any issue in the theoretical claims.

---

> ### Author Rebuttal · Authors · 2025-04-01
>
> We thank the reviewer for the constructive comments and valuable suggestions. We provide detailed responses to each comment in the following.
> ### Claims And Evidence
> **C1 Detailed ablation study**:
>
> **C1-Ans**: We include AUROC results in the ablation study table. To show full combinatorial results and enhance the readability, we also reformatted our ablation study table. Note that RSP (Residual Space Projection) and DCM (Dynamic Covariance Modeling) must be built on top of the DME (Dynamic Matrix Estimation) module. DCM is introduced in Section 4.3.
> |DME|RSP|DCM|CIFAR10 DenseNet (AUROC/FPR)|CIFAR100 DenseNet (AUROC/FPR) |ImageNet ViT (AUROC/FPR)| ImageNet ResNet50 (AUROC/FPR)|
> |-|-|-|-|-|-|-|
> |||| 95.19/18.86| 89.47/35.84| 92.51/33.16| 82.49/60.43|
> |✓||| 95.47/17.51| 88.05/36.61|92.65/32.31| 81.46/61.77|
> |✓|✓|| 94.08/23.92| 85.69/53.70|82.72/86.32|87.31/52.17|
> |✓|| ✓| 96.24/16.38|92.17/30.16|94.24/27.05|81.88/60.49|
> |✓| ✓|✓|**96.83**/**14.63**|**92.38**/**29.98**|**94.27**/**26.94**|**87.43**/**51.77**|
>
> As discussed in Section 5.5, the large between-class covariance distorts the geometry with which the data can not form a dense manifold. Thus, DCM is essential on some backbones, like DenseNet. We also discussed this in detail in Section C of the appendix. As shown in the table, the DME can solely improve the performance on  CIFAR10-DenseNet and ViT. Also, the RSP is essential in ResNet50.
> ### Methods And Evaluation Criteria:
> **C2 Experiments on near-OOD detection and iNaturelist**:
>
> **C2-Ans**:We have reported iNaturelist results in Tab. 9 of Appendix E. The near-OOD results can be viewed in **Reviewer YqZR's C4**
> ### Experimental Designs Or Analyses:
> **C3 Include NECO and VIM in CIFAR benchmark (Tab. 1) and DINO (Tab. 3)**:
>
> **C3-Ans**: We provide the experimental results of NECO and VIM on CIFAR benchmark and the results of Residual score on DINO in the following.
> || CIFAR10 DenseNet (AUROC/FPR) | CIFAR100 DenseNet (AUROC/FPR) |
> |-|-|-|
> |VIM| 93.44/33.66| 87.95/51.39|
> |NECO| 94.51/27.92| 79.79/72.43|
> |Ours| **96.83**/**14.63**| **92.38**/**29.98**|
>
> ||DINO(AUROC/FPR)|
> |-|-|
> |Residual (VIM)| 88.25/51.36|
> |Ours| **91.65**/**38.23**|
>
> Since DINO does not have the linear classifier as the supervised pretrained model does, for the DINO experiment, we only evaluate Residual score and discard the energy score in VIM.
>
> ### Relation To Broader Scientific Literature:
> **C4**: Detailed differences and relevance of the proposed approach compared to VIM and NECO
>
> **C4-Ans**: Both methods(VIM, NECO) only consider subspace information without direction-specific adjustments, potentially ignoring part of the ID information geometry in the distance-based score. Additionally, these methods do not adaptively utilize test-time information to correct for possible distortions in the training distribution, further limiting their sensitivity to novel or shifted feature directions. We will revise this discussion in Section 3 of the manuscript. The VIM and NECO can be interpreted as the matrix-induced distance-based scores with the formula $d_M(f) = \sqrt{(f-f_a)M(f-f_a)^\top }$. From this perspective, NECO and VIM can be explained as these methods to find better geometry of the distance for OOD detection. However, these methods ignore the fact that the matrix $M$ or geometry may be affected by the outliers in ID distribution (Details can be viewed in R2(YqZR)'s C6(Q2)). In our proposed approach, $M$ is related to real time features $f$, denoted as $\mathcal{M}(f)$. Specifically, $\mathcal{M}(f) = (Cov-r_f^\top r_f)^{-1}$, where $Cov$ is the covariance matrix from available features in the train set and $r_f$ is the projected real time features $f$ projected on residual space.  For VIM and NECO, $M = Res(Cov)$ and $M=Pri(Cov)$, with $Res(\cdot)$ and $Pri(\cdot)$ being the residual and principal spaces, respectively. This indicates that $M$ remains the same for both methods, regardless of any real-time features. In addition, they may ignore part of the ID information geometry in the distance-based score.
> ### Other Strengths And Weaknesses
> **C5**: The name of sub-section 2.2
>
> **C5-Ans**: We have updated the subtitle of section 2.2 to Post-hoc OOD detection methods.
>
> **C6**: The parameters of the OOD criterion are fixed for any given test-sample.
>
> **C6-Ans**: The dynamic component aims to mitigate covariance distortion caused by outliers in real-time features, specifically in Mahalanobis distance. Please view **R2(YqZR)'s C6(Q2)** for clearer motivation for our method. Our method relies on the initial computation of $\Sigma_R$ and $B$ as necessary initialization steps, while the dynamic component is provided by real-time recalibration of the covariance structure individually tailored for each test sample, significantly enhancing adaptability and robustness.
> ### Other Comments Or Suggestions:
> **C7**: Typos
>
> **C7-Ans**:  Thank you for pointing out the typos in our paper. The z in s(z) should be f. We have revised these typos

---

> > ### Comment · Reviewer_WNVA · 2025-04-07
> >
> > I thank the authors for their rebuttal. I read their answers to my and other reviewers' concerns, and I am mostly satisfied.
> >
> > However, I wonder why the proposed method shows such different behaviors between the (ImageNet ResNet50) and the other columns in the ablation. The DME + RSP combination seems detrimental in almost all configurations, and only the DME + DCM set-up brings consistent gains. This tendency is being reversed for R-50, but there is no explanation for it.

---

> > > ### Author Response · Authors · 2025-04-08
> > >
> > > Thank you for your response. Our full model achieves the best performance across all four different settings, highlighting the complementary nature of **DCM** and **RSP**.
> > >
> > > Since OOD data can also lie in the space between two ID clusters, the between-class covariance matrix may not only reflect ID information. Therefore, it is crucial to emphasize within-class covariance in OOD detection. To address this, **DCM** transforms the original embedding space, where clusters from different classes are separated, into a center-aligned embedding space by leveraging the within-class covariance matrix.
> > >
> > > Here (https://i.imgur.com/L9Mjnhj.png), we simulate two subspaces (i.e., well-clustered and poorly clustered space) to demonstrate the differing impacts of applying **DCM**. Comparing the upper and lower figures illustrates that the more clustered the embedding space is, the greater the improvement when using **DCM**. In practice, ImageNet-ViT has a better-structured embedding space compared to ImageNet-ResNet50, as indicated by the top-1 classification error.
> > >
> > > **RSP** leverages the residual space to highlight the covariance matrix dimensions that are highly affected by the ID outliers, so that these dimensions can be dynamically adjusted with incoming samples. For a well-clustered subspace, **RSP** can be detrimental in this situation due to the large between-class covariance. But it can solely work on poorly clustered scenarios since within-class covariance dominates the overall covariance matrix.  Notably, in our ablation study, without **DCM**, **RSP** can only be computed using the full covariance matrix（within-class + between-class), **differing from the **RSP** in the full model**, which specifically optimizes the within-class covariance.
> > >
> > >
> > >
> > > **Why is the tendency on ResNet50 different**: As discussed above, if the embedding space exhibits low inter-class separability, **DCM** does not significantly alter the feature distribution, and between-class covariance minimally impacts the overall covariance. In the following, we compare the largest eigenvalue of residual space projection matrix with and without between-class covariance to illustrate this phenomenon. From the table, we observe a significantly smaller difference in the largest eigenvalue for ResNet50 compared to ViT. This is why **DCM** can hardly improve the performance on ImageNet ResNet50, but solely **RSP** can improve the performance a lot on ResNet50.
> > >
> > > ||ImageNet ViT(w./w.o.)|ImageNet ResNet50(w./w.o.)|
> > > |-|-|-|
> > > |largest eigenvalue|1.2$e^{-3}$/5.7$e^{-4}$|8.2$e^{-6}$/7.6$e^{-6}$|

---

### Decision · Program_Chairs · 2025-05-01

**Decision:**

Accept (poster)

**Comment:**

In the intial review round, this paper received two Weak Accepts and one Weak Reject. Reviewers' opinions were diverse. They recognized several strengths, including novelty (YqZR, 6R2e), effectiveness of the method (YqZR, 6R2e), theoretical basis (YqZR), and clarity of the paper (YqZR). However, they also raised several concerns, such as computational time (YqZR), lack of experiments and comparisons (WNVA, YqZR), lack of analysis (WNVA, YqZR, 6R2e), and lack of novelty (WNVA).

The authors submitted a rebuttal, which was acknowledged by all reviewers. The authors did an excellent job and successfully addressed the concerns mentioned above. In particular, Reviewer WNVA initially questioned the novelty based on the overlap with ViM and NECO. In their rebuttal, the authors emphasized the use of test time inputs. Reviewer WNVA was convinced by the rebuttal and updated the score to Weak Accept.

Reviewer 6R2e was concerned about the small effectiveness of the projection when using ViT. The authors tried to justify this on the grounds that the ViT latent space as a whole already well separates IDs and OODs. However, given that ViT is the standard for classification and other tasks, this may be a limitation that should be discussed. This is also supported by the fact in the ablation study that introducing the projection (RSP) alone does not improve performance, except when ResNet50 is used. In a related matter, when further combined with DCM, the results suddenly improve, which is somewhat difficult to interpret. The authors are encouraged to discuss of these points in the final version.